# Pay Attention to CTC: Fast and Robust Pseudo-Labelling for Unified Speech Recognition

**Alexandros Haliassos**[1,2]**, Rodrigo Mira**[2]**, Stavros Petridis**[1,2]
[1]NatWest AI Research    [2]Imperial College London
alexandros.haliassos@natwest.com

## Abstract

Unified Speech Recognition (USR) has emerged as a semi-supervised framework for training a single model for audio, visual, and audiovisual speech recognition, achieving state-of-the-art results on in-distribution benchmarks. However, its reliance on autoregressive pseudo-labelling makes training expensive, while its decoupled supervision of CTC and attention branches increases susceptibility to self-reinforcing errors, particularly under distribution shifts involving longer sequences, noise, or unseen domains. We propose CTC-driven teacher forcing, where greedily decoded CTC pseudo-labels are fed into the decoder to generate attention targets in a single forward pass. Although these can be globally incoherent, in the pseudo-labelling setting they enable efficient and effective knowledge transfer. Because CTC and CTC-driven attention pseudo-labels have the same length, the decoder can predict both simultaneously, benefiting from the robustness of CTC and the expressiveness of attention without costly beam search. We further propose mixed sampling to mitigate the exposure bias of the decoder relying solely on CTC inputs. The resulting method, USR 2.0, halves training time, improves robustness to out-of-distribution inputs, and achieves state-of-the-art results on LRS3, LRS2, and WildVSR, surpassing USR and modality-specific self-supervised baselines.

## 1 Introduction

Speech recognition includes multiple closely related tasks: automatic speech recognition (ASR) from audio (Abdel-Hamid et al., 2014; Chiu et al., 2018), visual speech recognition (VSR) from lip movements (Assael et al., 2016; Martinez et al., 2020), and audiovisual speech recognition (AVSR), which combines both modalities (Petridis et al., 2018; Ma et al., 2021). Despite their complementary nature, ASR, VSR, and AVSR have traditionally been studied in isolation (Baevski et al., 2020; Hsu et al., 2021; Assael et al., 2016), leading to separate models that are not only more cumbersome to deploy, but also forgo the advantages of sharing representations across modalities (Akbari et al., 2021; Girdhar et al., 2022). Recent work in audiovisual self-supervised learning (Shi et al., 2022a; Zhu et al., 2023; Lian et al., 2023) has explored unified pre-training across modalities but still often relies on separate fine-tuning stages for ASR, VSR, and AVSR, resulting in distinct models per task.

Unified Speech Recognition (USR) (Haliassos et al., 2024a) offers a promising step toward unification by incorporating unlabelled data during fine-tuning, thus reducing modality-specific sensitivity and enabling a single model to achieve state-of-the-art in-distribution performance across all tasks. Its decoupled two-branch architecture combines Connectionist Temporal Classification (CTC) (Graves et al., 2006) and attention-based (Chorowski et al., 2015) supervision, where each student branch is independently guided by pseudo-labels (PLs) from the corresponding branch of a momentum teacher (Tarvainen & Valpola, 2017). While effective in distribution, this design leaves USR's attention decoder with two key limitations: (1) PL generation is a bottleneck, as its autoregressive (AR) decoding is slow and must run at every training step, and (2) decoupled supervision increases vulnerability to out-of-distribution (OOD) errors. That is, mistakes from AR decoding are compounded by the self-reinforcing training loop (Sohn et al., 2020): the student can be harmed by noisy AR PLs from the teacher, which can in turn degrade the teacher itself as it is a moving average of the student.

To address these limitations, we propose USR 2.0, which combines the efficiency and robustness of CTC with the expressiveness of attention-based decoding. As shown in Figure 1, CTC decoding is fast

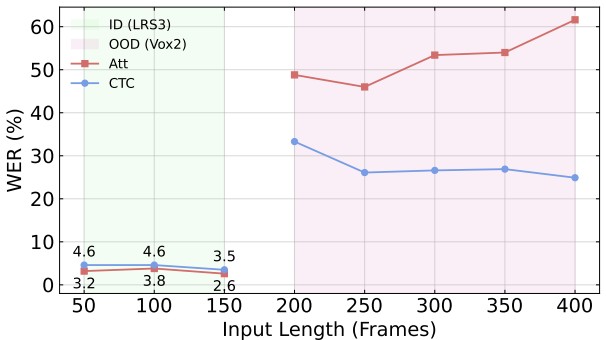 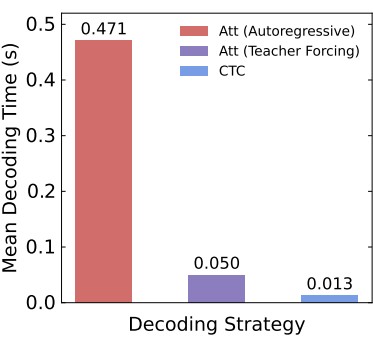

Figure 1: **CTC vs. attention-based decoding (greedy).** Left: AVSR word error rate (WER) of USR on in- (LRS3) and out-of-distribution (VoxCeleb2, automatically transcribed) samples. CTC decoding is notably more robust to domain shift and long sequences, while autoregressive attention-based decoding performs best in-distribution. Right: Decoding speed on a H200 GPU. CTC is ∼40× faster than autoregressive decoding. While teacher forcing could significantly speed up attention-based decoding, it typically relies on ground-truth tokens unavailable during pseudo-labelling.

and robust to domain shift due to monotonic alignment and conditional independence assumptions, while attention-based decoding has higher in-distribution quality but depends on slow AR generation. USR 2.0's core idea is CTC-driven teacher forcing: instead of decoding the teacher autoregressively, we feed its greedily decoded CTC outputs into the decoder to generate attention-based PLs in a single forward pass, removing the AR bottleneck. While this choice may appear counterintuitive—since the outputs can lack global coherence—a key insight is that in a pseudo-labelling setting coherence is unnecessary: teacher and student operate under the same forced inputs, making knowledge transfer effective. Further, because these attention-based PLs are derived from the CTC PLs (and are hence aligned), the student can predict both in a single decoder forward pass, coupling the two branches and improving robustness to OOD samples. Finally, as CTC-driven teacher forcing introduces a train–test mismatch (CTC inputs during training vs. AR tokens at inference), we mitigate it with a simple mixed sampling strategy that intermittently reintroduces AR decoding.

USR 2.0 improves robustness under domain shift, with large gains on long utterances, noisy audio, and cross-dataset evaluations across LibriSpeech (Panayotov et al., 2015), WildVSR (Djilali et al., 2024), and AVSpeech (Ephrat et al., 2018). It is also markedly less sensitive to beam size than USR, maintaining strong performance even under greedy decoding. These robustness improvements further translate to better *in-distribution* performance, as unlabelled data often comes from OOD sources (Chung et al., 2018; Shi et al., 2022a). As a result, USR 2.0 achieves state-of-the-art WER in various semi-supervised settings across ASR, VSR, and AVSR, *with nearly 2× faster training than USR*. Enabled by our findings, we scale USR 2.0 to a Huge model trained on ∼2500 hours of unlabelled data, yielding WERs of 17.6% (VSR), 0.9% (ASR), and 0.8% (AVSR) on LRS3, along with state-of-the-art results on LRS2 and WildVSR (Appendix C.1), all using a single unified model.

## 2 RELATED WORK

**CTC, attention, and joint training.** Speech recognition typically relies on CTC (Graves et al., 2006; Baevski et al., 2020; Hsu et al., 2021) or attention-based encoder–decoder models (Chorowski et al., 2015; Chan et al., 2016; Radford et al., 2023). CTC enables efficient training and inference via monotonic alignment and conditional independence, improving robustness in noisy or out-of-distribution settings (Kim et al., 2017), whereas attention-based models capture richer dependencies through autoregressive decoding but may suffer from alignment errors and degrade under distribution shift (Watanabe et al., 2017). Joint CTC–attention training (Kim et al., 2017) supervises a shared encoder via separate branches, with inference performed by joint beam search and CTC rescoring (see Appendix B). While this strategy has been adopted in VSR and AVSR (Ma et al., 2022; 2023; Haliassos et al., 2022a; 2024a), its reliance on costly beam search limits its use in iterative pseudo-labelling. Non-autoregressive transformers (Ghazvininejad et al., 2019; Tian et al., 2020; Song et al., 2021) avoid AR decoding via parallel token generation but typically reduce accuracy, whereas our

method applies CTC guidance to the attention-based decoder only during pseudo-labelling—where output sequence coherence is unnecessary—while retaining accurate AR beam search at inference.

**Self-supervised audiovisual speech models.** Audiovisual self-supervised learning has shown promise for speech recognition by learning contextualised representations from large-scale unlabelled audiovisual data (Shi et al., 2022a; Zhu et al., 2023; Lian et al., 2023). Pre-training typically relies on masked prediction (Baevski et al., 2022; He et al., 2022) or audiovisual correspondence (Chung & Zisserman, 2017; Morgado et al., 2021), followed by fine-tuning with a CTC layer and/or attention-based decoder (Cappellazzo et al., 2024; Rouditchenko et al., 2024). Although recent methods unify modalities in a single encoder during pre-training (Shi et al., 2022a; Zhu et al., 2023), the sensitivity of supervised fine-tuning to task-specific hyperparameters often complicates joint optimisation across modalities (Hsu & Shi, 2022; Shi et al., 2022a). As a result, fine-tuning is typically performed separately for ASR, VSR, and AVSR, increasing deployment costs (Haliassos et al., 2022a; 2024b) and forgoing the potential benefits of shared representations across modalities (Akbari et al., 2021).

**Pseudo-labelling for speech recognition.** Pseudo-labelling is an effective strategy for semi-supervised speech recognition, where model-generated transcriptions for unlabelled data act as additional supervision (Sohn et al., 2020). Offline pseudo-labelling with a frozen model is efficient (Afouras et al., 2020; Ma et al., 2023), but requires external models and does not support iterative refinement. In contrast, self-training methods update pseudo-labels during training using the model itself, often with beam search (Park et al., 2020; Xu et al., 2020; Kahn et al., 2020). However, beam search is computationally expensive, especially at scale, prompting some works to adopt CTC-only supervision (Likhomanenko et al., 2020; Higuchi et al., 2021), at the cost of reduced sequence modelling capacity (Rouditchenko et al., 2023). USR (Haliassos et al., 2024a) shows that fine-tuning with unlabelled data using greedy CTC and attention-based pseudo-labelling mitigates the task-specific sensitivity observed in self-supervised approaches, enabling a *single* model for ASR, VSR, and AVSR. Building on this, we revisit USR's decoupled pseudo-labelling scheme to improve training efficiency, robustness to distribution shifts, and in-distribution performance.

# 3 BACKGROUND: USR

USR (Haliassos et al., 2024a) is a semi-supervised student-teacher framework for training a single model to perform ASR, VSR, and AVSR. It uses a shared encoder with two output heads–a CTC layer and an attention-based decoder–and leverages unlabelled data via iterative pseudo-labelling (see Figure 2, left). The student receives masked audio, video, and audiovisual inputs, and is trained to match the teacher's outputs on unmasked audiovisual inputs, while also learning from ground-truth labels where available. Key components of USR are outlined below (more details in Appendix B.2).

**Teacher–student setup.** Let $\theta_S$ and $\theta_T$ be the parameters of the student and teacher models. The student is optimised via gradient descent, and the teacher is updated as an exponential moving average of the student via $\theta_T \leftarrow \tau\theta_T + (1-\tau)\theta_S$, where $\tau$ follows a cosine schedule from 0.998 to 1.

**Pseudo-labelling.** Given an unlabelled audiovisual speech input represented as a feature sequence of length $L$, the teacher generates two types of pseudo-labels (PLs) via greedy decoding. The CTC head produces frame-level PLs $\tilde{y}_t^{\text{CTC}}$ for $t \in [1, L]$, and the decoder (attention branch) autoregressively produces token-level PLs $\tilde{y}_u^{\text{Att}}$ for $u \in [1, U_{\text{AR}}]$, where $U_{\text{AR}}$ is the decoder output length:

$$\tilde{y}_t^{\text{CTC}} = \arg\max_{y_t} P_{\text{CTC}}(y_t \mid x; \theta_T), \quad \tilde{y}_u^{\text{Att}} = \arg\max_{y_u} P_{\text{Att}}(y_u \mid \tilde{y}_{<u}^{\text{Att}}, x; \theta_T). \tag{1}$$

**Loss on unlabelled data.** The student is trained to match the teacher's PLs for all masked modalities. Let $\hat{y}^{\omega,m}$ denote the student's output from head $\omega \in \{\text{CTC}, \text{Att}\}$ for modality $m \in \{\text{A}, \text{V}, \text{AV}\}$. USR uses the cross-entropy loss (frame-wise or token-wise depending on the head $\omega$):

$$\mathcal{L}^{\omega,m} = \text{CE}(\hat{y}^{\omega,m}, \tilde{y}^{\omega}), \tag{2}$$

where we omit time and token indices $t$ and $u$ for ease of notation. The total unlabelled loss is a weighted combination across heads and modalities, and is jointly optimised with the supervised loss.

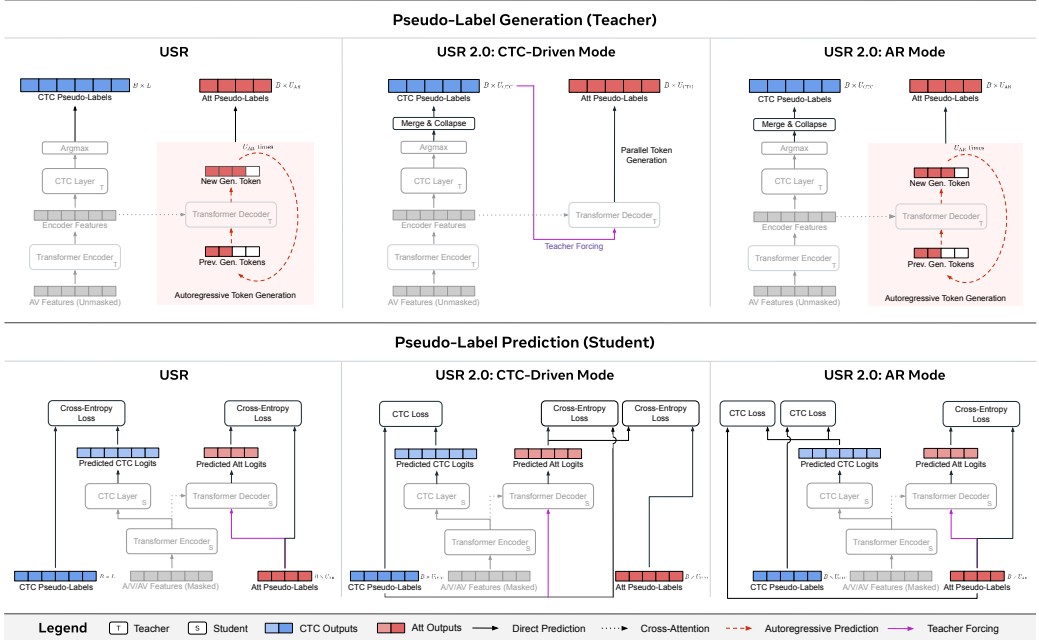

Figure 2: **Pseudo-labelling in USR and USR 2.0** In **USR** (left), the teacher generates CTC and attention-based pseudo-labels (PLs) from unmasked audiovisual inputs: CTC PLs are generated in parallel, while attention-based PLs require autoregressive decoding. The student predicts each target type independently, leading to decoupled supervision. **USR 2.0** introduces two modes for tighter integration. In CTC-driven mode (centre), attention-based PLs are generated by feeding collapsed CTC PLs into the decoder via teacher forcing, avoiding autoregression. The student decoder predicts both types of PLs. In AR mode (right), the teacher operates autoregressively as in USR, and the student's CTC branch predicts both CTC and attention-based PLs. USR 2.0 alternates between modes at each iteration: CTC-driven mode improves efficiency and robustness, while AR mode mitigates exposure bias. *Note: $B$ refers to batch size, and typically, $U_{CTC}, U_{AR} \ll L$ (not to scale in the figure).*

**Limitations.** We argue that a key limitation of USR is its decoupled supervision: the student's CTC and decoder branches are trained *independently* using their respective PLs. This design makes the attention decoder brittle on OOD inputs (e.g., noisy audio or long utterances), because (1) greedy autoregressive (AR) decoding can fail and produce cascading errors (Kim et al., 2017; Fan et al., 2018), and (2) errors are reinforced in the training loop, as noisy AR PLs supervise the student, which can then degrade the teacher via EMA updates. We validate this effect empirically in Section 5.

Equally problematic is the computational cost of generating attention-based PLs: because they are produced autoregressively, one forward pass is required per output token. Despite key–value caching, this makes pseudo-labelling slow, especially since decoding is performed at every training step. This inefficiency becomes a major bottleneck for scaling to larger models and longer sequences, and it limits the practical utility of semi-supervised training on large unlabelled corpora.

## 4 PROPOSED METHOD

To overcome these limitations, we propose USR 2.0 for faster and more robust pseudo-labelling. The key idea is CTC-driven teacher forcing, where greedily decoded CTC PLs replace slow AR decoding to generate attention targets in a single forward pass. This design both removes the AR bottleneck and transfers the robustness of CTC to the decoder via joint CTC-attention PL prediction. To address the resulting train–test discrepancy, we propose a simple mixed sampling strategy, intermittently reintroducing AR decoding. Together, these components preserve the modelling flexibility of attention while improving efficiency and robustness under distribution shift. An overview is in Figure 2.

## 4.1 CTC-Driven Teacher Forcing

We propose to leverage the teacher's CTC outputs to guide the generation of attention-based PLs. Given an unlabelled audiovisual input $x$, we first greedily decode the teacher's CTC head, then apply a merge-and-collapse operation by removing blank tokens and merging repeated symbols:

$$\tilde{y}_t = \arg\max_{y_t} P_{\text{CTC}}(y_t \mid x; \theta_T), \quad \tilde{y}^{\text{CTC}} = \text{collapse}(\tilde{y}_{1:L}) \tag{3}$$

where $\text{collapse}(\cdot)$ denotes the standard CTC post-processing operation (Graves et al., 2006). This produces a sequence $\tilde{y}^{\text{CTC}}$ of length $U_{\text{CTC}} \ll L$, which we then feed as input to the teacher decoder:

$$\tilde{y}_u^{\text{Att}} = \arg\max_{y_u} P_{\text{Att}}(y_u \mid \tilde{y}_{<u}^{\text{CTC}}, x; \theta_T), \quad u = 1, \ldots, U_{\text{CTC}}, \tag{4}$$

replacing the autoregressive conditioning on past attention outputs with the *fixed* CTC token prefix $\tilde{y}_{<u}^{\text{CTC}}$. This removes the sequential dependency, allowing the entire sequence of attention-based PLs to be generated in *parallel* with a single forward pass. As a result, the modelling strength of attention-based pseudo-labelling is preserved while avoiding the high cost and fragility of autoregression.

**Aligned targets.**    Since the attention-based PLs are generated by conditioning the decoder directly on the CTC pseudo-labels, they inherit the same sequence length $U_{\text{CTC}}$. This shared alignment ensures that both pseudo-label types correspond position-wise, allowing the student decoder to predict them simultaneously in a single forward pass under teacher forcing on the CTC PLs. This design lets the decoder inherit robustness from predicting CTC pseudo-labels while retaining flexibility from predicting attention pseudo-labels.

**Global coherence.**    In CTC-driven teacher forcing, each decoder output $\tilde{y}_u^{\text{Att}}$ is conditioned on the fixed prefix of CTC tokens $\tilde{y}_{<u}^{\text{CTC}}$ rather than past decoder outputs $\tilde{y}_{<u}^{\text{Att}}$. A natural concern is that the resulting sequence $\tilde{y}^{\text{Att}}$ may lack coherence (see Appendix C.4 for discussion). While this prevents its use for *inference-time* decoding, the student decoder learns effectively during *self-training* with token-wise cross-entropy because of matched conditioning: teacher and student are conditioned on the same CTC-derived sequence, which is itself coherent. The student is trained to predict the teacher decoder's most likely next token under this shared input. The student decoder therefore learns a stable mapping from a coherent CTC prefix to the teacher's conditionally valid next-token prediction. Because inference is autoregressive and relies on this learned prefix-to-next-token relation, global incoherence of the teacher-generated sequence does not hinder learning.

## 4.2 Mixed Sampling

While CTC-driven teacher forcing allows attention-based PLs to be generated efficiently, it introduces a mismatch: the decoder is trained using inputs derived from the teacher's CTC predictions, whereas at inference time it autoregressively conditions on its own past outputs. This discrepancy can be seen as a form of "exposure bias," which can hurt performance (Ranzato et al., 2015). In addition to this input-side mismatch, CTC's conditional-independence assumptions may also occasionally yield imperfect CTC predictions on challenging, in-distribution segments, and these errors can propagate through the teacher to produce weaker attention-based pseudo-labels. To address these issues, we introduce a simple yet effective mixed sampling strategy, similar to scheduled sampling (Bengio et al., 2015) but for a different type of exposure bias. At each training step, we randomly sample a mode: with probability 0.5 we use the CTC-driven mode, otherwise we adopt the standard AR mode[1].

In **CTC-driven mode**, the student CTC branch outputs $\hat{y}^{\text{CTC},m}$ for modality $m$, while the decoder outputs $\hat{y}^{\text{Att},m}$, conditioned on the same CTC PLs $\tilde{y}^{\text{CTC}}$ used to condition the teacher. The losses are:

$$\mathcal{L}^{\text{CTC},m} = \text{CTC}(\hat{y}^{\text{CTC},m}, \tilde{y}^{\text{CTC}}), \quad \mathcal{L}^{\text{Att},m} = 0.5\,\text{CE}(\hat{y}^{\text{Att},m}, \tilde{y}^{\text{Att}}) + 0.5\,\text{CE}(\hat{y}^{\text{Att},m}, \tilde{y}^{\text{CTC}}). \tag{5}$$

The decoder is supervised with both the teacher's attention-based pseudo-labels $\tilde{y}^{\text{Att}}$ and the CTC targets $\tilde{y}^{\text{CTC}}$ to leverage the strengths of both PL types. The student's CTC branch is supervised only with $\tilde{y}^{\text{CTC}}$, since $\tilde{y}^{\text{Att}}$ may lack coherence (as explained) and could degrade CTC training.

---

[1]We also experimented with an adaptive sampling schedule (Appendix C.2), where the probability of AR pseudo-labelling increases over training. This strategy performed similarly to the fixed 0.5 probability used in USR 2.0, so we adopt the simpler approach in the main method.

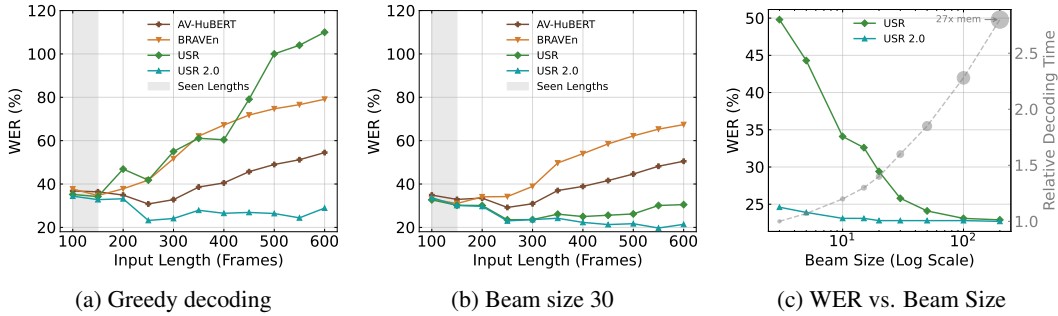

(a) Greedy decoding        (b) Beam size 30        (c) WER vs. Beam Size

Figure 3: **Robustness to long utterances.** (a) Greedy decoding: USR 2.0 maintains robustness to longer input lengths, significantly outperforming other models. (b) Beam search (beam size $= 30$, joint CTC-attention decoding) improves USR robustness but still lags behind USR 2.0. (c) Increasing beam size reduces the WER gap between USR and USR 2.0, but at a significant computational and memory cost. The size of the markers corresponds to the relative memory cost with batched beam search.

In **AR mode**, the attention PLs $\tilde{y}^{\text{Att}}$ are generated via standard AR decoding to alleviate train-test mismatch. The student decoder is trained to predict $\tilde{y}^{\text{Att}}$, while the CTC branch is jointly supervised with both $\tilde{y}^{\text{CTC}}$ and $\tilde{y}^{\text{Att}}$, as both now represent coherent sequences. The corresponding losses are:

$$\mathcal{L}^{\text{CTC},m} = 0.5\,\text{CTC}(\hat{y}^{\text{CTC},m}, \tilde{y}^{\text{CTC}}) + 0.5\,\text{CTC}(\hat{y}^{\text{CTC},m}, \tilde{y}^{\text{Att}}), \quad \mathcal{L}^{\text{Att},m} = \text{CE}(\hat{y}^{\text{Att},m}, \tilde{y}^{\text{Att}}). \quad (6)$$

Note that only the CTC branch receives both types of pseudo-labels as supervision: $\tilde{y}^{\text{CTC}}$ and $\tilde{y}^{\text{Att}}$ may differ in length, making it infeasible to supervise the decoder on both in a single forward pass.

## 4.3 IMPLEMENTATION DETAILS

Our implementation follows USR, with the main difference being our proposed pseudo-labelling strategy. We summarise the setup below; full details, including model sizes and hyperparameters, are given in Appendix A.

**Architecture.** We use a transformer encoder–decoder with modality-specific ResNet-18 backbones (He et al., 2016; Stafylakis & Tzimiropoulos, 2017) for audio and video. The audio encoder subsamples to match the visual frame rate (25 fps). Audio and visual features are projected via linear layers into a shared transformer dimension, while audiovisual features are formed by concatenation and applying a separate projection (Haliassos et al., 2024a). The resulting representations are processed by a transformer encoder with relative positional embeddings (Dai et al., 2019). We experiment with four model sizes (Base, Base+, Large, Huge); details are in Table 5.

**Loss weights.** We follow joint CTC–attention training (Kim et al., 2017), combining the CTC loss (weight 0.1) with attention-based cross-entropy loss using label smoothing 0.1 (Szegedy et al., 2016). Following USR, we use modality-specific weights of 0.3 (visual) and 0.7 (audio and audiovisual), and unlabelled-to-labelled loss ratios of 0.97 (visual) and 0.75 (audio and audiovisual).

**Thresholding.** We adopt confidence-based filtering as in USR, with a threshold of 0.8. For collapsed CTC PLs, sequence-level confidence is computed as the average log-probability over token predictions.

**Inference.** Unless otherwise stated, inference uses joint CTC–attention decoding via ESPnet (Watanabe et al., 2018), with beam size 40 and CTC weight 0.1. We use a 1,000-token SentencePiece vocabulary (Kudo & Richardson, 2018) trained on the labelled data.

## 5 OUT-OF-DISTRIBUTION RESULTS

Out-of-distribution robustness is critical not only for deployment but also for semi-supervised learning, since unlabelled data often comes from domains outside the labelled distribution. Better

Table 1: **Comparison of WER (%) across different SNR levels (dB)** for ASR and AVSR on the LRS3 test set (beam size 30). None of the methods saw noise-augmented samples during training.

| Method | All Samples | | | | | Samples with > 100 Frames | | | | |
|---|---|---|---|---|---|---|---|---|---|---|
| | 10 dB | 5 dB | 0 dB | -5 dB | Avg | 10 dB | 5 dB | 0 dB | -5 dB | Avg |
| **ASR** | | | | | | | | | | |
| AV-HuBERT | 8.4 | 18.9 | 54.2 | 94.6 | 44.0 | 6.5 | 16.2 | 49.3 | **97.2** | 42.3 |
| BRAVEn | 7.2 | 16.8 | 50.1 | 99.2 | 43.3 | 6.4 | 13.8 | 49.2 | 108.4 | 44.5 |
| USR | 5.8 | 14.3 | 48.5 | 104.4 | 43.3 | 4.9 | 12.6 | 48.1 | 111.7 | 44.3 |
| USR 2.0 | **5.2** | **13.4** | **44.0** | **94.4** | **39.3** | **3.8** | **10.6** | **42.8** | 98.3 | **38.9** |
| **AVSR** | | | | | | | | | | |
| AV-HuBERT | 5.8 | 10.5 | 24.3 | 56.9 | 24.4 | 4.4 | 8.0 | 18.0 | 48.7 | 19.8 |
| BRAVEn | 7.2 | 14.0 | 32.2 | 54.6 | 27.0 | 6.5 | 12.7 | 30.9 | 55.5 | 26.4 |
| USR | 4.0 | 6.4 | 14.8 | 36.5 | 15.4 | 2.7 | 4.6 | 11.6 | 29.1 | 12.0 |
| USR 2.0 | **3.7** | **5.6** | **14.0** | **33.1** | **14.1** | **2.6** | **4.3** | **10.4** | **26.0** | **10.8** |

robustness yields higher-quality pseudo-labels and thus more reliable training. To assess USR 2.0's robustness, we evaluate the Base model under long utterances, noisy audio, and unseen datasets, comparing against USR and strong self-supervised baselines AV-HuBERT (Shi et al., 2022a) and BRAVEn (Haliassos et al., 2024b). All experiments are conducted in the low-resource setting (Shi et al., 2022a), where the labelled set is the 30-hour "trainval" partition of LRS3 (Afouras et al., 2018) and the remaining LRS3 samples are treated as unlabelled data.

## 5.1 ROBUSTNESS TO LONG UTTERANCES

We first evaluate robustness to long sequences using ∼2,000 automatically transcribed samples from VoxCeleb2 (Chung et al., 2018), treating Whisper (Radford et al., 2023), trained on far more data, as an oracle for WER. The test set is bucketed into ∼200-sample groups by input length (50–600 frames, in 50-frame increments). The longest labelled training utterance is 155 frames, while unlabelled samples reach 600, so inputs beyond 155 represent a distributional shift relative to the labelled data.

**Greedy decoding.** Figure 3a shows WER versus input length under greedy (attention-based) decoding, which is important for pseudo-labelling. All methods perform similarly for inputs under 150 frames, but WER rises sharply beyond that for BRAVEn, AV-HuBERT, and especially USR, likely due to compounded autoregressive errors. While longer sequences should theoretically help by providing more context, distributional mismatch dominates. In contrast, USR 2.0 remains stable due to its integrated CTC and attention supervision, which mitigates autoregressive drift.

**Beam search.** We next evaluate joint CTC–attention decoding with beam size 30. For AV-HuBERT, which lacks CTC, we use attention-only decoding. As shown in Figure 3b, USR benefits strongly from beam search, as CTC scoring helps correct decoder errors. However, performance still drops beyond 400 frames, suggesting that inference-time correction does not fully compensate for a poorly trained decoder. In contrast, USR 2.0 remains strong, indicating greater underlying robustness.

**Beam size.** Finally, we vary the beam size and plot WER for USR and USR 2.0 across all samples (Figure 3c). At small beams, USR 2.0 clearly outperforms USR, highlighting its robustness under fast decoding. As beam size increases, the gap narrows, with very large beams compensating for USR's weaker decoder at the cost of significantly higher latency and memory. Overall, USR 2.0 performs far better with small beams, making it well suited for pseudo-labelling and low-latency applications.

## 5.2 ROBUSTNESS TO NOISE

In Table 1, we evaluate generalisation to noise on LRS3 with additive babble noise from NOI-SEX (Varga & Steeneken, 1993) at SNRs from 10 dB to –5 dB. As most LRS3 test samples are short, we also report results on those longer than 100 frames to better reflect real-world inputs. Importantly, none of the models were trained with noise augmentation, making this a zero-shot evaluation.

Table 2: **In-distribution comparisons with self-supervised methods.** LRS3 WER (%) results for low- (left) and high-resource (right) labelled data settings. Best results in **bold**. We specify model size and pre-training dataset for each setting in **bold**. "Shared params" denotes whether a unified model is used. *Trained on LRS2+LRS3 (labelled) and English-only VoxCeleb2+AVSpeech (unlabelled).

| Method | Shared params | V | A | AV | Method | Shared params | V | A | AV |
|---|---|---|---|---|---|---|---|---|---|
| **Base, LRS3** | | | | | **Base+, LRS3+Vox2** | | | | |
| AV-HuBERT | ✗ | 51.8 | 4.9 | 4.7 | AV-HuBERT | ✗ | 34.8 | 2.0 | 1.8 |
| RAVEn | ✗ | 47.0 | 4.7 | - | VATLM | ✗ | 34.2 | - | 1.7 |
| AV-data2vec | ✗ | 45.2 | 4.4 | 4.2 | RAVEn | ✗ | 33.1 | 1.9 | - |
| BRAVEn | ✗ | 43.4 | 4.0 | 4.0 | AV-data2vec | ✗ | 32.9 | 1.7 | 1.4 |
| USR | ✓ | **36.0** | 3.2 | 3.0 | Lip2Vec | ✗ | 34.1 | - | - |
| USR 2.0 | ✓ | 36.2 | **3.0** | **2.9** | BRAVEn | ✗ | 28.8 | **1.4** | - |
| | | | | | USR | ✓ | 26.5 | 1.6 | 1.3 |
| **Base+, LRS3+Vox2** | | | | | USR 2.0 | ✓ | **24.8** | **1.4** | **1.2** |
| AV-HuBERT | ✗ | 46.1 | 4.6 | 4.0 | | | | | |
| RAVEn | ✗ | 40.2 | 3.8 | - | **Large, LRS3+Vox2** | | | | |
| AV-data2vec | ✗ | 37.8 | 3.7 | 3.3 | AV-HuBERT | ✗ | 28.6 | 1.3 | 1.4 |
| BRAVEn | ✗ | 35.1 | 3.0 | - | VATLM | ✗ | 28.4 | - | 1.2 |
| USR | ✓ | 28.4 | 2.6 | 2.5 | RAVEn | ✗ | 28.2 | 1.4 | - |
| USR 2.0 | ✓ | **26.4** | **2.5** | **2.4** | AV-data2vec | ✗ | 28.5 | 1.3 | 1.3 |
| | | | | | Lip2Vec | ✗ | 26.0 | - | - |
| **Large, LRS3+Vox2** | | | | | BRAVEn | ✗ | 26.6 | **1.2** | - |
| AV-HuBERT | ✗ | 32.5 | 2.9 | 3.3 | u-HuBERT | ✓ | 29.1 | 1.5 | 1.3 |
| RAVEn | ✗ | 32.5 | 2.7 | - | USR | ✓ | 22.3 | **1.2** | 1.1 |
| AV-data2vec | ✗ | 30.8 | 2.7 | 2.7 | USR 2.0 | ✓ | **21.5** | 1.3 | **1.0** |
| BRAVEn | ✗ | 30.8 | **2.3** | - | | | | | |
| USR | ✓ | 26.9 | 2.4 | 2.4 | **Huge*** | | | | |
| USR 2.0 | ✓ | **23.7** | **2.3** | **2.2** | USR 2.0 | ✓ | 17.6 | 0.9 | 0.8 |

For ASR, USR 2.0 clearly outperforms all baselines at moderate noise levels (10 to 0 dB), showing strong robustness. At -5 dB SNR, where the audio distribution shifts substantially, its performance is comparable to AV-HuBERT. On longer samples, the relative gap between USR 2.0 and USR slightly widens, suggesting that USR 2.0's decoder may be more resilient to accumulated errors over time.

Consistent with prior works (Ma et al., 2021; Haliassos et al., 2024b;a), AVSR outperforms ASR under noise by leveraging visual information. Across all SNRs, the pseudo-labelling methods (USR and USR 2.0) significantly outperform the self-supervised baselines (BRAVEn and AV-HuBERT). Interestingly, the improvement of USR 2.0 over USR remains stable across sequence lengths, suggesting that its AVSR gains are less dependent on input duration than in ASR.

## 5.3 Robustness to OOD Datasets

We evaluate robustness to OOD datasets with substantial gaps between training and test distributions, including shifts in vocabulary, speaker accents, background noise, and recording conditions. We use LibriSpeech test-clean (Panayotov et al., 2015) for ASR, WildVSR (Djilali et al., 2024) for VSR, and 1,000 manually filtered AVSpeech (Ephrat et al., 2018) samples for AVSR, transcribed using Whisper (Radford et al., 2023). As shown in Table 3, USR 2.0 outperforms all baselines by a wide margin under greedy decoding across all tasks, highlighting the robustness of its improved decoder to diverse, real-world shifts.

Table 3: **Performance on OOD datasets**: LibriSpeech (LibriS), Wild-VSR, and AVSpeech (AVS). We report WER (%) under greedy decoding.

| Method | LibriS | WildVSR | AVS |
|---|---|---|---|
| AV-HuBERT | 29.1 | 82.4 | 26.0 |
| BRAVEn | 38.4 | 81.2 | 44.6 |
| USR | 25.3 | 80.0 | 34.7 |
| USR 2.0 | **15.4** | **73.7** | **25.0** |

## 6 In-Distribution Results

We now evaluate performance on the LRS3 benchmark, which serves as an *in-distribution* setting where the labelled and test data come from the same domain. However, because the unlabelled data often comes from other sources (e.g., VoxCeleb2), pseudo-labelling robustness remains critical.

Table 4: **AVSR WER (%) for different pseudo-label types** for each branch in CTC-driven and AR modes. Default settings are in ~~gray~~.

| CTC Head Preds | | Decoder Preds | | ID | OOD |
|---|---|---|---|---|---|
| CTC PL | Att PL | CTC PL | Att PL | | |
| *CTC-Driven Mode* | | | | | |
| ✓ | ✗ | ✓ | ✓ | **3.2** | **24.2** |
| ✓ | ✗ | ✗ | ✓ | 3.3 | 35.1 |
| ✓ | ✓ | ✓ | ✓ | **3.2** | 24.3 |
| ✓ | ✗ | ✓ | ✗ | 3.6 | 25.5 |
| *AR Mode* | | | | | |
| ✓ | ✓ | ✗ | ✓ | 2.9 | **40.1** |
| ✓ | ✗ | ✗ | ✓ | 3.0 | 45.1 |
| ✗ | ✓ | ✗ | ✓ | **2.8** | 52.3 |

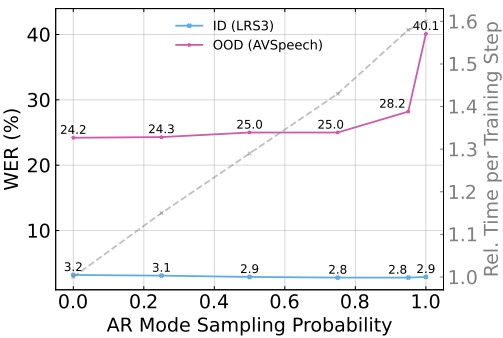

Figure 4: **AVSR WER (%) vs. AR mode sampling probability** on in-distribution (LRS3) and out-of-distribution (AVSpeech) samples.

**Low- and high-resource settings.** Table 2 presents WERs for low- (30h) and high-resource (433h) settings (Shi et al., 2022a), comparing against self-supervised methods, including RAVEn (Haliassos et al., 2022a), AV-data2vec (Lian et al., 2023), VATLM (Zhu et al., 2023), Lip2Vec (Djilali et al., 2023), and u-HuBERT (Hsu & Shi, 2022). USR 2.0 matches or outperforms the state of the art, even methods that train separate models per task. Gains over USR are more pronounced with VoxCeleb2 pre-training, particularly for VSR, which relies more heavily on pseudo-labels. These results support our hypothesis that improving pseudo-labelling benefits even in-distribution performance.

**Training efficiency.** Figure 5 plots WER for VSR against wall-clock training time for USR and USR 2.0 across different model scales and pre-training datasets. We observe approximately 2× faster training for each setting, driven by two key factors: (i) faster training steps due to CTC-driven teacher forcing; and (ii) faster convergence, requiring fewer epochs (50 vs. 75; see Appendix C.5). These gains make USR 2.0 substantially more efficient for semi-supervised training at scale.

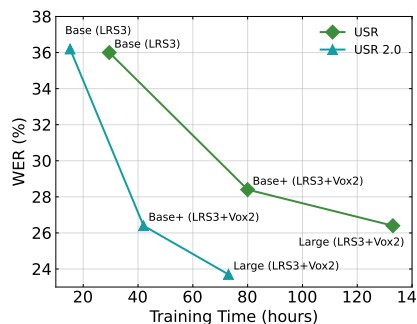

**Scaling.** Building on these results, we scale USR 2.0 to a Huge model trained on the combined LRS2 (Son Chung et al., 2017) and LRS3 labelled sets, along with a large unlabelled corpus from English-only VoxCeleb2 and AVSpeech. The model achieves WERs of 17.6%, 0.9%, and 0.8% on VSR, ASR, and AVSR, respectively, demonstrating USR 2.0's scalability. Comparisons with additional methods are in Appendix C.1.

Figure 5: **VSR WER vs. training time** for the low-resource setting.

# 7 ABLATIONS

We ablate the choice of pseudo-label (PL) targets for each branch under the two modes: CTC-driven and AR. To isolate their effects, we disable AR mode when evaluating CTC-driven mode, and vice versa. All experiments use the low-resource setup with the Base model (see Section 5). We report AVSR WER on both in-distribution (ID; LRS3 test) and out-of-distribution (OOD; AVSpeech) data. More ablations are given in Appendix C.2.

**CTC-driven mode (Table 4).** By default, the CTC head predicts CTC PLs, while the decoder predicts both CTC and attention-based PLs. Removing CTC supervision from the decoder harms OOD performance, confirming its value for robustness. Dropping attention-based targets reduces ID performance, underscoring their importance for sequence modelling. Thus, both are crucial. Interestingly, supervising the CTC head with attention-based PLs does not noticeably degrade performance, suggesting that AR inconsistencies may be offset by higher-quality conditional predictions.

Finally, conditioning the decoder on CTC pseudo-labels in this mode avoids conditioning solely on AR-generated pseudo-labels, which can be error-prone under domain shift due to cascaded autoregressive mistakes. This effect is reflected in the large OOD gap between rows 2 of "CTC-Driven Mode" (35.1%) and "AR Mode" (45.1%).

**AR mode (Table 4).**   Here, the default assigns only attention-based PLs to the decoder (due to the target length mismatch) and both PL types to the CTC head. Removing attention-based targets from the CTC head reduces performance. Using only attention-based PLs in both branches degrades OOD performance, reaffirming the role of CTC PLs for robustness. Overall, ID and OOD trends are largely uncorrelated, and AR performs significantly worse than CTC-driven mode on OOD data.

**Effect of mixed sampling probability (Figure 4).**   We evaluate the effect of varying the probability of sampling the AR mode during training. As this probability increases, ID performance improves, likely due to reduced exposure bias. OOD performance remains relatively stable across most of the range but degrades sharply when approaching AR-only training. Training time per iteration also increases steadily with higher AR probability due to the cost of AR decoding. The sampling probability therefore controls the balance between ID accuracy, OOD robustness, and training efficiency, and may be tuned to match application needs, with our default of 0.5 offering a strong overall balance.

## 8    CONCLUSION

We presented USR 2.0, which addresses two key limitations of USR that prevent scaling to larger models and diverse unlabelled data: the computational cost of autoregressive pseudo-labelling and the decoupled design that reduces robustness to out-of-distribution inputs. By introducing CTC-driven teacher forcing and mixed sampling, USR 2.0 combines the *efficiency* and *robustness* of CTC with the *modelling flexibility* of attention-based decoding. This design improves generalisation to long utterances, noisy conditions, and unseen datasets, while reducing training time by nearly half. These robustness gains also translate to improved in-distribution performance, since unlabelled data often comes from out-of-distribution sources. Scaling both model and corpus, USR 2.0 achieves state-of-the-art performance across ASR, VSR, and AVSR with a *single* model. Beyond USR, our insights can be applied to other speech recognition tasks, including audio-only, streaming, and multilingual ASR, which are active research areas where large unlabelled datasets are common. More broadly, the CTC-driven teacher-forcing paradigm could be applied to other sequence-to-sequence settings where the input and output follow the same temporal order but lack explicit frame-level alignment, such as handwriting recognition, music transcription, or DNA/protein sequencing. We hope our insights will prove useful for future work on scalable sequence-to-sequence self-training beyond speech.

### ACKNOWLEDGEMENTS

Data collection, processing, and experiments were conducted at Imperial College London.

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

# A EXPERIMENTAL SETUP

## A.1 DATASETS

**LRS3.** The LRS3 dataset (Afouras et al., 2018) consists of approximately 430 hours of transcribed TED Talk videos, featuring thousands of speakers and an open vocabulary. Videos are automatically segmented into sentence-level utterances, with audio and visual streams synchronised via facial tracking. The dataset exhibits rich variability in lighting, speaking styles, and head poses. The test set, used for evaluation only, contains around 1 hour of speech by speakers held out from training. LRS3 is released under the Creative Commons BY-NC-ND 4.0 license.

**LRS2.** Sourced from British television broadcasts, LRS2 (Son Chung et al., 2017) provides 223 hours of audiovisual content with sentence-level transcriptions. While similar in structure to LRS3, the video content in LRS2 is more controlled, with generally better lighting and resolution. The test set contains roughly 30 minutes of clips spoken by unseen speakers and is frequently used for benchmarking visual and audiovisual speech models. It is available for academic, non-commercial research use.

**WildVSR.** WildVSR (Djilali et al., 2024) is a challenging VSR benchmark collected from open-domain video content. Designed to test generalisation, it includes more extreme conditions than LRS3 or LRS2, such as diverse ethnicities, head motions, and video resolutions. It comprises around 5 hours of test material without overlap with LRS3 and is used exclusively for evaluation. WildVSR is licensed under CC BY-NC-ND 4.0.

**AVSpeech.** AVSpeech (Ephrat et al., 2018) contains over 4,700 hours of face-aligned speech segments harvested from over 150,000 YouTube videos. Its scale and speaker diversity make it valuable for self- or semi-supervised learning. We follow prior work (Ma et al., 2023) in filtering the dataset to retain 1,323 hours of only English-speaking videos. AVSpeech is licensed under the Creative Commons Attribution 4.0 International License (CC BY 4.0).

**VoxCeleb2.** VoxCeleb2 (Chung et al., 2018) comprises over 2,400 hours of unconstrained, in-the-wild recordings from 6,000+ speakers. The data spans multiple languages and noisy environments such as interviews and live performances. We use an English-only subset curated in prior work (Shi et al., 2022a), which includes 1,326 hours of utterances. VoxCeleb2 is distributed under the CC BY-NC-ND 4.0 license.

**LibriSpeech.** We include the test-clean set of LibriSpeech (Panayotov et al., 2015) as an additional audio-only evaluation benchmark. It consists of 5.4 hours of read speech from audiobook recordings, featuring high-quality audio with minimal background noise. LibriSpeech is released under the CC BY 4.0 license.

The lists of sampled videos used for out-of-distribution evaluation on AVSpeech and VoxCeleb2 are included in the `README.md` of the source code provided in the supplementary material.

## A.2 EXTERNAL CODEBASES

For the robustness experiments in Section 5, we directly ran the official codebases of prior methods to ensure fair and consistent comparisons. Specifically, we used:

- AV-HuBERT (Shi et al., 2022a): `https://github.com/facebookresearch/av_hubert`
- BRAVEn (Haliassos et al., 2024b): `https://github.com/ahaliassos/raven`
- USR (Haliassos et al., 2024a): `https://github.com/ahaliassos/usr`

Where available, we used the officially released pre-trained models. In cases where no suitable model was provided (i.e., audiovisual BRAVEn, audio-only AV-HuBERT, and audiovisual AV-HuBERT without noise augmentations), we trained the models ourselves using the official hyperparameters and training protocols.

Table 5: **Model hyperparameters.** Parameter count includes encoder, decoder, and feature extractors.

|  | Base | Base+ | Large | Huge |
|---|---|---|---|---|
| Total parameters (M) | 86 | 171 | 503 | 953 |
| Transformer blocks (enc / dec) | 12 / 6 | 12 / 6 | 24 / 9 | 36 / 9 |
| Attention dim | 512 | 768 | 1024 | 1280 |
| MLP size | 2048 | 3072 | 4096 | 5120 |
| Heads | 8 | 12 | 16 | 16 |

Table 6: **Training hyperparameters** for different pre-training datasets and model sizes. "All" refers to the combination of LRS2, LRS3, VoxCeleb2, and AVSpeech. Additional hyperparameters are provided in Appendix A.5.

| Hyperparameter | LRS3 (Base) | LRS3+VoxCeleb2 | All (Huge) |
|---|---|---|---|
| Learning rate | 3e-3 | 2e-3 | 7e-4 |
| Weight decay | 0.04 | 0.04 | 0.04 |
| $(\beta_1, \beta_2)$ | $(0.9, 0.98)$ | $(0.9, 0.98)$ | $(0.9, 0.98)$ |
| Drop path rate | 0.1 | 0.1 (Base), 0.2 (Large) | 0.3 |
| Frames per GPU (lab./unlab.) | 600 / 4400 | 300 / 3200 | – |
| Frames per GPU (high-resource) | – | 1200 / 3200 | 700 / 2400 |

For all other baselines and state-of-the-art comparisons, we report numbers as provided in the original publications.

## A.3 PRE-PROCESSING

Video frames are first stabilised to reduce jitter, then cropped to a $96 \times 96$ region centred on the speaker's mouth, and finally converted to grayscale. This pipeline follows prior work (Shi et al., 2022a;b; Haliassos et al., 2022a; 2024b;a). No pre-processing is applied to the raw audio.

## A.4 MODEL VARIANTS

We employ four model sizes: Base, Base+, Large, and Huge. Each variant shares the same feature extractors, with the encoder and decoder size being the primary differentiating factors. Configurations are summarised in Table 5. Note that Base+ corresponds to the Base model used in related works (Shi et al., 2022a; Lian et al., 2023; Zhu et al., 2023).

## A.5 TRAINING HYPERPARAMETERS

**Optimiser and scheduling.** Models are trained for 50 epochs using AdamW (Loshchilov & Hutter, 2017), with a 15-epoch linear warmup (Goyal et al., 2017) followed by cosine decay (Loshchilov & Hutter, 2016). Drop path regularisation (Huang et al., 2016) and gradient clipping (threshold 3.0) are applied for stability.

**Augmentations.** Video augmentations include random cropping ($88 \times 88$) and horizontal flipping with probability 0.5, applied consistently across frames. Both modalities also use temporal zero-masking for the student model to enforce context understanding: video inputs are masked with max duration 0.4s, audio with 0.6s, following Ma et al. (2022); Haliassos et al. (2022a). Teacher inputs are unmasked to preserve PL quality.

**Training resources.** We fix seed to 42. Training takes ∼1 day for Base (8 H200 GPUs), ∼2 days for Base+ (32 GPUs), and ∼3 days for Large (32 GPUs). Base uses LRS3, Base+ and Large use LRS3+VoxCeleb2, and Huge uses LRS2+LRS3+VoxCeleb2+AVSpeech over ∼4 days on 64 GPUs. Frame counts per GPU are tuned for utilisation.

**Initialisation.**  All models are initialised from the same self-supervised checkpoints as USR. For Huge, we pre-train using the Large USR hyperparameters from Haliassos et al. (2024a) before semi-supervised fine-tuning.

**Summary.**  Table 6 lists key hyperparameters. Training configs, dataset prep, and evaluation code are included in the supplementary material.

# B  SUPPLEMENTARY METHODOLOGICAL BACKGROUND

## B.1  CTC AND ATTENTION-BASED MODELS

**Connectionist Temporal Classification (CTC).**  CTC (Graves et al., 2006) models the probability of a target sequence $y = (y_1, \ldots, y_U)$ given input frames $x$ by marginalising over all monotonic alignment paths $\pi$:

$$P_{\text{CTC}}(y \mid x) = \sum_{\pi \in \mathcal{B}^{-1}(y)} \prod_{t=1}^{T} P(\pi_t \mid x), \tag{7}$$

where $\pi_t$ is the label (or blank) at timestep $t$ and $\mathcal{B}$ collapses repeats and blanks to recover $y$. This formulation (i) enforces a left-to-right monotonic alignment, since paths are built only by inserting blanks or repeats (never by reordering labels), and (ii) assumes each $\pi_t$ is conditionally independent of $\pi_{<t}$ given $x$, thereby allowing the path probability to factorise as $\prod_t P(\pi_t \mid x)$. This explicit alignment mechanism tends to be robust under distribution shifts, but the conditional independence assumption limits modelling capabilities (Watanabe et al., 2017).

**Attention-based encoder–decoder models.**  In contrast, attention-based encoder-decoder models define the conditional probability:

$$P_{\text{Att}}(y \mid x) = \prod_{u=1}^{U} P(y_u \mid y_{<u}, x). \tag{8}$$

Unlike CTC, attention-based decoders do not assume conditional independence across timesteps, allowing them to model long-range dependencies and complex token interactions. However, they do not enforce monotonic alignment and must instead learn alignment patterns purely from data. As a result, they are highly expressive but often brittle under distribution shifts, such as longer input sequences or noisy data, where learned alignments may fail to generalise.

During training, these models are typically trained using teacher forcing (Williams & Zipser, 1989), where the ground-truth tokens $y_{<u}$ are fed as inputs to the decoder instead of its own predictions. This stabilises learning and significantly accelerates computation by enabling parallel sequence processing in transformer-based architectures. However, at inference time, the model must operate autoregressively, meaning each predicted token must be fed back into the decoder for the next step.

**Joint CTC–attention training and decoding.**  To leverage the strengths of both CTC and attention-based models, hybrid architectures incorporate a joint CTC–attention framework (Watanabe et al., 2017). This approach optimises a multi-task objective:

$$\mathcal{L} = \lambda \mathcal{L}_{\text{CTC}} + (1 - \lambda)\mathcal{L}_{\text{Att}}, \tag{9}$$

where $\mathcal{L}_{\text{CTC}}$ is the CTC loss (typically applied to a projection of the transformer encoder outputs), $\mathcal{L}_{\text{Att}}$ is the attention-based cross-entropy loss (applied to the decoder outputs), and $\lambda$ is a tunable hyperparameter balancing the two objectives.

At inference time, joint decoding is often performed using beam search, combining CTC and attention scores:

$$S(y') = \alpha \log P_{\text{CTC}}(y' \mid x) + (1 - \alpha) \log P_{\text{Att}}(y' \mid x), \tag{10}$$

where $y'$ is a candidate hypothesis and $\alpha$ controls the relative contribution of the CTC branch. The CTC probabilities help constrain the beam, reducing alignment errors in attention-based decoding and improving robustness to distribution shifts.

While effective, beam search is computationally expensive and thus impractical in iterative self-training setups, where decoding must be performed repeatedly over large unlabelled datasets.

## B.2 ADDITIONAL DETAILS ON USR

While the main paper focuses on our new pseudo-labelling generation and prediction strategy, this section supplements the brief recap in Section 3 with components of USR that are not the focus in USR 2.0, specifically the confidence filtering strategy (excluded from the main text for simplicity) and a more detailed formulation of the semi-supervised loss. For clarity, we describe the setting using one labelled and one unlabelled sample per iteration (with audio, visual, and audiovisual views); in practice, batches of $b_l$ labelled and $b_u$ unlabelled samples are used, typically with $b_u > b_l$.

**Shared parameters.** USR is a single model trained to perform ASR, VSR, and AVSR jointly. Aside from modality-specific feature extractors (audio and visual frontends), the remaining components (encoder and decoder heads) are shared across modalities. These shared components operate on token sequences, allowing audio, visual, and audiovisual inputs to be processed in a unified transformer architecture after appropriate modality-specific projections. A key benefit of this design in the self-training setup is that pseudo-labels only need to be generated once per unlabelled audiovisual sample. These shared pseudo-labels can then supervise all three modality-specific inputs (audio-only, video-only, and audiovisual), amortising the cost of pseudo-label generation.

**Confidence filtering.** To reduce the impact of erroneous pseudo-labels during self-training, USR applies confidence-based filtering, discarding low-confidence predictions from the teacher model. Let $p^{\text{CTC}} \in [0,1]^{L \times |\mathcal{V}|}$ and $p^{\text{Att}} \in [0,1]^{U_{\text{AR}} \times |\mathcal{V}|}$ denote the softmax outputs of the teacher's CTC and attention-based heads, respectively, where $\mathcal{V}$ is the subword vocabulary. Frame- and token-wise masks are obtained as follows:

$$r_t^{\text{CTC}} = \mathbb{1}(\max_k p_{t,k}^{\text{CTC}} \geq \tau), \qquad r_u^{\text{Att}} = \mathbb{1}(\max_k p_{u,k}^{\text{Att}} \geq \tau), \tag{11}$$

with confidence threshold $\tau = 0.8$.

In USR 2.0, the CTC PLs are token-level (rather than frame-level) due to the collapse operation, so it is difficult to use the frame-wise confidence scores $c_t^{\text{CTC}} = \max_k p_{t,k}^{\text{CTC}}$ directly in the CTC loss function. Instead, we compute a single sequence-level confidence:

$$\text{conf}(\tilde{y}^{\text{CTC}}) = \exp\left(\frac{1}{L} \sum_{t=1}^{L} \log c_t^{\text{CTC}}\right). \tag{12}$$

The CTC sequence-level mask is then given by

$$r^{\text{CTC}} = \mathbb{1}\left(\text{conf}(\tilde{y}^{\text{CTC}}) \geq \tau\right). \tag{13}$$

This reflects the fact that in USR 2.0, CTC PL filtering must occur at the sequence level due to the way the CTC loss is computed.

**Unlabelled losses.** Let $\hat{y}^{\omega,m}$ denote the student predictions from head $\omega \in \{\text{CTC}, \text{Att}\}$ for modality $m \in \{\text{A, V, AV}\}$. In USR, the cross-entropy losses on unlabelled data (with confidence masking) are

$$\mathcal{L}_u^{\text{CTC},m} = \frac{1}{L} \sum_{t=1}^{L} r_t^{\text{CTC}} \text{CE}(\hat{y}_t^{\text{CTC},m}, \tilde{y}_t^{\text{CTC}}), \tag{14}$$

$$\mathcal{L}_u^{\text{Att},m} = \frac{1}{U_{\text{AR}}} \sum_{u=1}^{U_{\text{AR}}} r_u^{\text{Att}} \text{CE}(\hat{y}_u^{\text{Att},m}, \tilde{y}_u^{\text{Att}}). \tag{15}$$

In USR 2.0, the unlabelled losses from equation 5–equation 6 are similarly masked using $r^{\text{CTC}}$ and $r_u^{\text{Att}}$.

The CTC and attention unlabelled losses are then combined per modality using:

$$\mathcal{L}_u^m = \lambda_{\text{CTC}} \mathcal{L}_u^{\text{CTC},m} + (1 - \lambda_{\text{CTC}}) \mathcal{L}_u^{\text{Att},m}, \quad \lambda_{\text{CTC}} \in [0,1]. \tag{16}$$

**Labelled losses.** For labelled samples, USR employs the standard joint CTC–attention objective described in Section B. The resulting per-modality labelled loss is denoted by $\mathcal{L}_l^m$.

Table 7: **Comparisons with the state-of-the-art on LRS3.** [*]Labels include automatic transcriptions from ASR models trained on large-scale, often non-public datasets. "Shared params" denotes whether a unified model is used at inference. "ST" denotes offline self-training. [†]Uses Whisper as its encoder, trained on ~680k hours of transcribed speech data (Radford et al., 2023).

| Method | Labelled hours | Unlabelled hours | Lang. model | Shared params | WER (%) V | A | AV |
|---|---|---|---|---|---|---|---|
| **Supervised**[*] | | | | | | | |
| V2P | 3,886 | - | ✗ | ✗ | 55.1 | - | - |
| RNN-T | 31,000 | - | ✗ | ✓ | 33.6 | 4.8 | 4.5 |
| VTP | 2,676 | - | ✓ | ✗ | 30.7 | - | - |
| Llama-AVSR[†] | 680,000 | - | ✓ | ✗ | 24.0 | 0.8 | **0.8** |
| Auto-AVSR | 1,902 | - | ✓ | ✗ | 23.5 | 1.0 | 1.0 |
| Auto-AVSR | 3,448 | - | ✓ | ✗ | 19.1 | 1.0 | 0.9 |
| ViT3D-CM | 90,000 | - | ✗ | ✗ | 17.0 | - | 1.6 |
| SynthVSR | 6,720 | - | ✓ | ✗ | 16.9 | - | - |
| LP Conf | 100,000 | - | ✗ | ✗ | **12.8** | - | 0.9 |
| Whisper-Flamingo[†] | 680,000 | - | ✓ | ✗ | - | **0.7** | 0.8 |
| **Self/semi-supervised** | | | | | | | |
| AV-HuBERT w/ ST | 433 | 1,326 | ✗ | ✗ | 28.6 | - | - |
| RAVEn w/ ST | 433 | 1,326 | ✓ | ✗ | 23.1 | 1.4 | - |
| BRAVEn w/ ST | 433 | 2,649 | ✓ | ✗ | 20.1 | 1.1 | - |
| USR | 433 | 1,326 | ✓ | ✓ | 21.5 | 1.2 | 1.1 |
| **USR 2.0** | 656 | 2,649 | ✗ | ✓ | **17.6** | **0.9** | **0.8** |

**Final semi-supervised objective.** Let $w_m \in [0,1]$ denote the per-modality weight and $\gamma_m \in [0,1]$ control the balance between unlabelled and labelled losses. The full loss is:

$$\mathcal{L}_{\text{semi}} = \sum_{m \in \{\text{A,V,AV}\}} w_m \left[ \gamma_m \, \mathcal{L}_u^m + (1 - \gamma_m) \, \mathcal{L}_l^m \right]. \tag{17}$$

As described in Section 4.3, we set:

$$w_{\text{A}} = w_{\text{AV}} = 0.7, \quad w_{\text{V}} = 0.3, \quad \gamma_{\text{A}} = \gamma_{\text{AV}} = 0.75, \quad \gamma_{\text{V}} = 0.97. \tag{18}$$

## C ADDITIONAL RESULTS

### C.1 FULL STATE-OF-THE-ART COMPARISONS

Tables 7, 8, and 9 present comparisons with prior work on LRS3, LRS2, and WildVSR. We report results from strong supervised systems trained with vastly more labelled or automatically transcribed data, including V2P (Shillingford et al., 2018), RNN-T (Makino et al., 2019), VTP (Prajwal et al., 2022), Llama-AVSR (Cappellazzo et al., 2024), Auto-AVSR (Ma et al., 2023), ViT3D-CM (Serdyuk et al., 2022), SynthVSR (Liu et al., 2023), LP Conf (Chang et al., 2024), Whisper-Flamingo (Rouditchenko et al., 2024), CM-seq2seq (Ma et al., 2021), and CM-aux (Ma et al., 2022). All results for USR 2.0 refer to our Huge model, trained on LRS2 and LRS3 labelled data and unlabelled English-only VoxCeleb2 and AVSpeech, *without* the use of an external language model.

USR 2.0 achieves strong performance across all modalities. On LRS2 and LRS3, it outperforms Auto-AVSR, which leverages automatic transcriptions from external ASR models, and is on par with Whisper-Flamingo and Llama-AVSR, which initialise from Whisper models trained on 680,000 hours of transcribed speech data. On WildVSR (Table 9), which contains video clips with high visual variability, USR 2.0 obtains state-of-the-art performance, surpassing prior heavily supervised systems. This highlights its robustness to in-the-wild conditions such as variable head pose, lighting, and background.

Table 8: **Comparisons with the state-of-the-art on LRS2.** *Labels include automatic transcriptions from ASR models trained on large-scale, often non-public datasets. "Shared params" denotes whether a unified model is used at inference. "ST" denotes offline self-training. †Uses Whisper as its encoder, trained on ~680k hours of transcribed speech data (Radford et al., 2023).

| Method | Labelled hours | Unlabelled hours | Lang. model | Shared params | WER (%) V | A | AV |
|---|---|---|---|---|---|---|---|
| **Supervised*** | | | | | | | |
| CM-seq2seq | 380 | - | ✓ | ✗ | 37.9 | 3.9 | 3.7 |
| CM-aux | 1,459 | - | ✓ | ✗ | 25.5 | - | - |
| VTP | 698 | - | ✓ | ✗ | 28.9 | - | - |
| VTP | 2,676 | - | ✓ | ✗ | 22.6 | - | - |
| Auto-AVSR | 818 | - | ✓ | ✗ | 27.9 | 2.6 | - |
| Auto-AVSR | 3,448 | - | ✓ | ✗ | **14.6** | 1.5 | 1.5 |
| Whisper-Flamingo† | 680,000 | - | ✓ | ✗ | - | **1.3** | **1.4** |
| **Self/semi-supervised** | | | | | | | |
| Uni-AVSR | 223 | 60,000 | ✗ | ✗ | 43.2 | 2.7 | 2.6 |
| LiRA | 223 | 433 | ✓ | ✗ | 38.8 | - | - |
| RAVEn | 223 | 1,759 | ✗ | ✗ | 23.2 | 2.5 | - |
| RAVEn w/ ST | 223 | 1,759 | ✓ | ✗ | 17.9 | 2.3 | - |
| USR | 223 | 1,759 | ✓ | ✓ | 15.4 | 1.9 | 1.9 |
| **USR 2.0** | 656 | 2,649 | ✗ | ✓ | **12.6** | **1.3** | **1.3** |

Table 9: **Comparisons with the state-of-the-art on WildVSR.** *Labels include automatic transcriptions from ASR models trained on large-scale, often non-public datasets. "Shared params" denotes whether a unified model is used at inference. "ST" denotes offline self-training.

| Method | Labelled hours | Unlabelled hours | Lang. model | Shared params | WER (%) |
|---|---|---|---|---|---|
| **Supervised*** | | | | | |
| CM-seq2seq | 1,459 | - | ✓ | ✗ | 58.4 |
| VTP | 698 | - | ✓ | ✗ | 75.6 |
| VTP | 2,676 | - | ✓ | ✗ | 68.7 |
| Auto-AVSR | 661 | - | ✓ | ✗ | 62.3 |
| Auto-AVSR | 1,759 | - | ✓ | ✗ | 49.3 |
| Auto-AVSR | 3,448 | - | ✓ | ✗ | **38.6** |
| **Self/semi-supervised** | | | | | |
| AV-HuBERT | 433 | 1,326 | ✗ | ✗ | 51.7 |
| AV-HuBERT w/ ST | 433 | 1,326 | ✓ | ✗ | 48.7 |
| RAVEn | 433 | 1,326 | ✗ | ✗ | 52.2 |
| RAVEn w/ ST | 433 | 1,326 | ✓ | ✗ | 46.7 |
| USR | 433 | 1,326 | ✓ | ✓ | 46.4 |
| **USR 2.0** | 656 | 2,649 | ✗ | ✓ | **38.5** |

Table 10: **Ablation results.** We report WER (%) for VSR (V), ASR (A), and AVSR (AV) on in-distribution test data (LRS3) and AVSR on out-of-distribution data (sampled, automatically transcribed AVSpeech). Default settings for USR 2.0 are highlighted in `gray`.

(a) Weight on attention vs. CTC pseudo-labels for decoder loss in CTC-driven mode.

| Weight | ID | | | OOD |
|---|---|---|---|---|
| | V | A | AV | AV |
| 0 | 36.7 | 3.7 | 3.6 | 25.5 |
| 0.25 | 37.1 | 3.5 | 3.4 | **24.2** |
| 0.5 | 36.6 | 3.3 | **3.2** | **24.2** |
| 0.75 | **36.4** | 3.2 | 3.2 | 28.1 |
| 1 | 36.5 | 3.3 | 3.3 | 35.1 |

(b) Weight on attention vs. CTC pseudo-labels for CTC loss in AR mode.

| Weight | ID | | | OOD |
|---|---|---|---|---|
| | V | A | AV | AV |
| 0 | 36.3 | 3.2 | 3.0 | 45.1 |
| 0.25 | 36.7 | 3.2 | 2.9 | 41.9 |
| 0.5 | 36.5 | 3.1 | 2.9 | **40.1** |
| 0.75 | 36.2 | **3.0** | 2.9 | 44.4 |
| 1 | **36.1** | **3.0** | **2.8** | 52.3 |

(c) Effect of CTC merge & collapse on pseudo-labels *in USR*.

| Method | ID | | | OOD |
|---|---|---|---|---|
| | V | A | AV | AV |
| Without | **36.0** | **3.2** | 3.0 | **34.7** |
| With | 36.2 | **3.2** | 2.9 | 36.9 |

(d) AR mode sampling schedule.

| Schedule | ID | | | OOD |
|---|---|---|---|---|
| | V | A | AV | AV |
| Constant | **36.2** | 3.0 | **2.9** | 25.0 |
| Linear | 36.5 | **2.9** | **2.9** | **24.8** |
| Cosine | 36.6 | 3.0 | **2.9** | 25.4 |

These results demonstrate the benefits of scaling both model capacity and unlabelled data within a unified semi-supervised learning framework. While USR 2.0 already outperforms many task-specific baselines, we anticipate further improvements through continued scaling of compute and data. Importantly, all results are obtained using a *single* shared model for ASR, VSR, and AVSR, without relying on task-specific tuning or external language models.

## C.2 ADDITIONAL ABLATIONS

We conduct additional ablations on the Base model in the low-resource setting, using LRS3 as the unlabelled dataset. We evaluate in-distribution performance on the LRS3 test set for VSR, ASR, and AVSR, and out-of-distribution performance on the sampled AVSpeech test set (AVSR with greedy decoding). Results are shown in Table 10.

**Loss weights for auxiliary pseudo-labels.** Recall that in CTC-driven mode, the decoder is trained to match both CTC and attention-based pseudo-labels. Similarly, in AR mode, the CTC branch is trained to predict both pseudo-label types. By default, the coefficients balancing the contribution of these losses are set to 0.5 in both cases (see equation 5–equation 6 in the main text). We ablate different values of these coefficients from $\{0, 0.25, 0.5, 0.75, 1\}$ for each prediction head.

As shown in Table 10(a), assigning greater weight to CTC pseudo-labels in the decoder loss generally improves out-of-distribution robustness but can degrade in-distribution accuracy, especially for ASR and AVSR, which are more sensitive to pseudo-label noise. This trade-off highlights the complementary inductive biases of CTC and attention: CTC promotes monotonic alignment and stability, while attention models long-range dependencies that are more beneficial in clean, in-distribution settings.

Interestingly, Table 10(b) shows that combining CTC and attention-based pseudo-labels in the CTC loss of the AR mode improves OOD performance over using either type alone, suggesting that the two supervision signals reinforce each other when used in the CTC branch.

**CTC merge & collapse.** USR uses a cross-entropy loss directly on frame-level pseudo-labels and predictions, without collapsing repeated tokens or removing blanks. In contrast, USR 2.0 applies the standard CTC post-processing step (merging repeats and deleting blanks) to convert frame-level outputs into token-level pseudo-labels. To isolate the effect of this change, we compare training USR

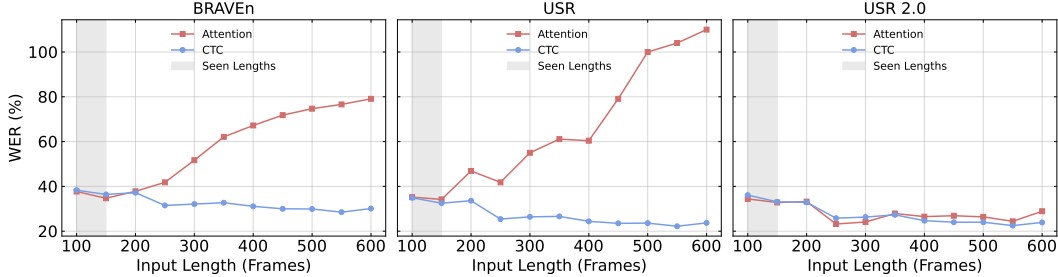

Figure 6: **WER vs. input length** on VoxCeleb2 (automatically transcribed using Whisper). Models are trained in the low-resource LRS3 setup, where labelled utterances are no longer than around 150 video frames. We evaluate CTC-only and attention-only greedy decoding for BRAVEn, USR, and USR 2.0.

with and without merge & collapse. As shown in Table 10(c), this choice has negligible impact on performance in USR. The benefit of collapsing in USR 2.0 stems not from improved label quality, but from enabling compatibility with teacher forcing in the decoder: it significantly reduces sequence length and maps the targets into the subword token space required by the decoder input.

**Sampling schedule.** USR 2.0 alternates between CTC-driven and AR modes when training on unlabelled data. By default, we use a constant sampling probability of $0.5$ for AR mode throughout training. A natural hypothesis is to begin with more frequent use of the more robust CTC-driven mode, then gradually increase the use of AR mode to improve expressiveness. To test this, we evaluate both linear and cosine schedules for the AR sampling probability, increasing from $0$ to $1$ over the course of training. As shown in Table 10(d), neither schedule outperforms the simpler constant strategy, suggesting that fixed mixing is sufficient in this setting.

### C.3 OUT-OF-DISTRIBUTION ROBUSTNESS

**Length generalisation of decoding branches.** Figure 6 evaluates the robustness of CTC-only and attention-only decoding across input lengths for BRAVEn, USR, and USR 2.0. We use the Base models trained in the low-resource setting (on LRS3), where the longest labelled utterances span roughly 150 frames. Evaluation is performed on out-of-distribution samples from VoxCeleb2, grouped into length buckets (e.g., 150–200, 200–250, etc), as in Figure 3 from the main text.

All models perform stably under CTC-only decoding, likely due to the robustness of monotonic alignment. However, attention-only decoding reveals notable differences. Both BRAVEn and USR exhibit sharp increases in WER as input length grows, reflecting the decoder's fragility on long, out-of-distribution utterances. Notably, USR deteriorates more severely than BRAVEn, likely due to its self-training loop, where autoregressive pseudo-labels are generated by a teacher that is itself derived from the student, compounding early decoding errors. In contrast, USR 2.0 maintains significantly better performance, suggesting that its training strategy (i.e., the transfer of alignment information from the CTC branch to the decoder) successfully stabilises attention-based predictions.

**Qualitative examples.** To illustrate this effect, Table 11 presents qualitative decoding examples on long, out-of-distribution utterances. Transformer decoders are known to struggle with such inputs under greedy decoding, often producing truncated or repetitive outputs (Fan et al., 2018; Holtzman et al., 2019). These issues are amplified in pseudo-labelling setups, where standard remedies like beam search with length penalties are too costly to apply. In USR, we observe such failures on utterances longer than those seen during supervised training, typically manifesting as omissions or degeneracies.

USR 2.0 mitigates these issues by jointly supervising the decoder with both CTC and attention-based pseudo-labels. This promotes more stable and complete sequence generation, even under greedy decoding. As shown in Table 11, USR 2.0 produces fluent hypotheses, while USR suffers from repetition and content loss. These findings echo our quantitative results in Figures 6 and 3 of the main text, where USR 2.0 maintains robustness on long utterances.

Table 11: **Qualitative analysis of USR vs. USR 2.0 on long, out-of-distribution utterances.** We show examples where USR either omits substantial segments (top) or degenerates into repetitions (bottom), both of which USR 2.0 resolves. Segments omitted by USR but correctly transcribed by USR 2.0 are highlighted in green; repetition errors by USR are highlighted in red.

| | **(a) Omission Failures** |
|---|---|
| GT | because with them we can break this cycle **of suffering without them the cycle will continue I'll leave you with the word that we use at the end** of every yoga class which also illustrates this fundamental truth I really believe in that we are all one |
| USR | because with them we can break this cycle of every yoga class which also illustrates this fundamental truth I really believe in is that we are all one |
| USR 2.0 | because with them we can break this cycle **of suffering without them the cycle will continue I'll leave you with the word that we use at the end** of every yoga class which also illustrates this fundamental truth I really believe in is that we are all one |
| GT | **so the evidence is there and it's going to be designed to interpret it correctly it will understand for example that books are very biased in the evidence they contain they only talk about kings and princes and elite white male people doing stuff** so it's a complicated problem but as it learns more about our objectives it will become more and more |
| USR | so it's a complicated problem but as it learns more about our objectives it will become more and more |
| USR 2.0 | **so the evidence is there it's going to be designed to interpret it correctly it will understand for example that books are very biased in the evidence they contain they only talk about kings and princes and elite white male people doing stuff** so it's a complicated problem but as it learns more about our objectives it will become more and more |
| GT | **you may find these stories inspirational I have to be perfectly honest I find them daunting a little bit unsettling because I look at my own life and I ask is this going to happen to me and if it does will I recognize it and if I recognize it will I have the guts to make the leap myself** so I took a look at some of these leaps into the void because if super successful peoples' success in fact does not hinge upon living on the edge |
| USR | so I took a look at some of these leaps into the void because if super successful people success in fact does not hinge upon living on the edge |
| USR 2.0 | **you may find these stories inspirational have to be perfectly honest I find them daunting a little bit unsettling because I look at my own life and I ask is this going to happen to me and if it does will I recognize it and if I recognize it will I have the guts to make the leap myself** so I took a look at some of these leaps into the void because if super successful people's success in fact does not hinge upon living on the edge |

| | **(b) Repetition Failures** |
|---|---|
| GT | here's the striking thing though here's what Shery says Shery says yeah it's an incredibly painful experience but she wouldn't have it any other way because what it's taught her is |
| USR | here's the striking thing though here's what Sharry says Sharry says **Sharry says Sharry says Sharry says Sharry says Sharry says Sharry says. . .** |
| USR 2.0 | here's the striking thing though here's what Shary says Shary says yes incredibly painful experience but she wouldn't have it any other way because it's what it's taught her is |
| GT | at the pebble on the bottom of the stream the stream is continuously moving and turbulent and that makes it very difficult to see the pebble on the bottom of the stream very much in the same way it's very difficult to see astronomical sources |
| USR | at the pubble on the bottom of stream the stream **the stream the stream the stream the stream the stream the stream the stream the stream. . .** |
| USR 2.0 | at the puble on the bottom of stream the stream is continuously moving and turbulent and that makes it very difficult to see the puble on the bottom of the stream very much in the same way it's very difficult to see astronomical sources |
| GT | your grandmother was on TEDx Youth and that was like so cool back then I want to go like I went to this and I did that and I saw this I want to tell them stories |
| USR | your grandmother was on TEDx Youth and that was like so cool back then see I want to go like I went to this and I did that and I saw this **and I saw this and I saw this and I saw this...** |
| USR 2.0 | your grandmother was on TEDx Youth and that was like so cool back then see I want to go like I went to this and I did that and I saw this and I wanted to tell them story |

**Example 1**

Ground-truth sentence    WE EXCISE IT FROM OUR BODIES

CTC pseudo-labels

CTC-driven attention
pseudo-labels

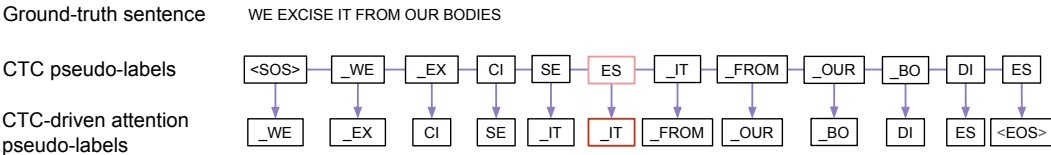

**Example 2**

Ground-truth sentence:    EVERYONE IS OKAY

CTC pseudo-labels

CTC-driven attention
pseudo-labels

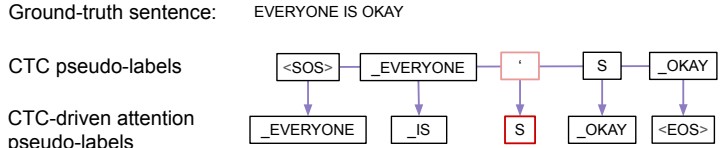

**Example 3**

Ground-truth sentence    I DO WANT TO TALK ABOUT IT

CTC pseudo-labels

CTC-driven attention
pseudo-labels

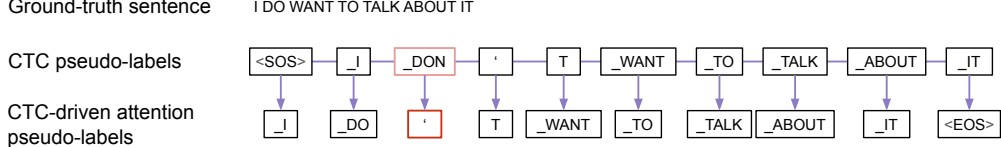

Figure 7: **Examples of AR inconsistencies** in decoder outputs under CTC-driven teacher forcing. In each case, the decoder is conditioned on CTC-derived prefixes, sometimes leading to repeated or malformed outputs at the sequence level, even though each token prediction is plausible given its input. These artefacts do not affect training, as the student learns to match the teacher's outputs under the same input.

## C.4    CTC-CONDITIONED SEQUENCE-LEVEL INCONSISTENCIES

When using CTC-driven teacher forcing to generate decoder targets for self-training, we observe that the resulting attention-based outputs may exhibit *sequence*-level inconsistencies, even though each individual token prediction is *conditionally* valid. Specifically, the decoder is conditioned on the sequence of subwords predicted by the CTC branch, and the decoder produces the most plausible next token given each CTC-derived prefix. If the decoder and CTC branches diverge on certain tokens, the resulting attention-based output may contain repeated or malformed phrases when interpreted as a complete sequence.

Figure 7 illustrates examples of this phenomenon. In the first case, the CTC output incorrectly includes an extra subword ("ES" after "EXCISE"). When conditioned on a prefix ending in "EXCISE", the decoder correctly predicts "IT". However, for the subsequent prediction, the decoder sees the CTC-based prefix ending in "EXCISEES" and again predicts "IT", which is a reasonable output under the given context. This results in a duplicated "IT" in the final sequence. In another case, the CTC branch outputs "EVERYONE'S", leading the decoder to generate "IS" when conditioned on the prefix "EVERYONE", followed by "S" under the prefix "EVERYONE'" , resulting in the malformed word "ISS". Finally, we show an example where the CTC output incorrectly predicts "DON'T" instead of "DO", prompting the decoder to generate "DO'T" due to the prefix "DON".

We emphasise again that these prefix-conditioned artefacts do *not* undermine the training process. The student decoder is trained to predict the teacher's outputs under the *same* CTC-driven inputs, making the decoder's predictions consistent *given* its inputs, even if they appear inconsistent at the sequence level.

Table 12: **WER (%) for USR 2.0 under different decoding strategies at inference.** We compare CTC-only decoding, attention-only greedy decoding, joint CTC + attention decoding with beam search (beam size 30), and CTC-driven teacher forcing

| Decoding Method | V | A | AV |
|---|---|---|---|
| CTC-only (greedy) | 45.6 | 4.1 | 4.0 |
| Attention-only (greedy) | 41.8 | 3.8 | 3.9 |
| CTC + attention (beam size 30) | **36.2** | **3.0** | **2.9** |
| CTC-driven teacher forcing | 47.1 | 4.2 | 4.1 |

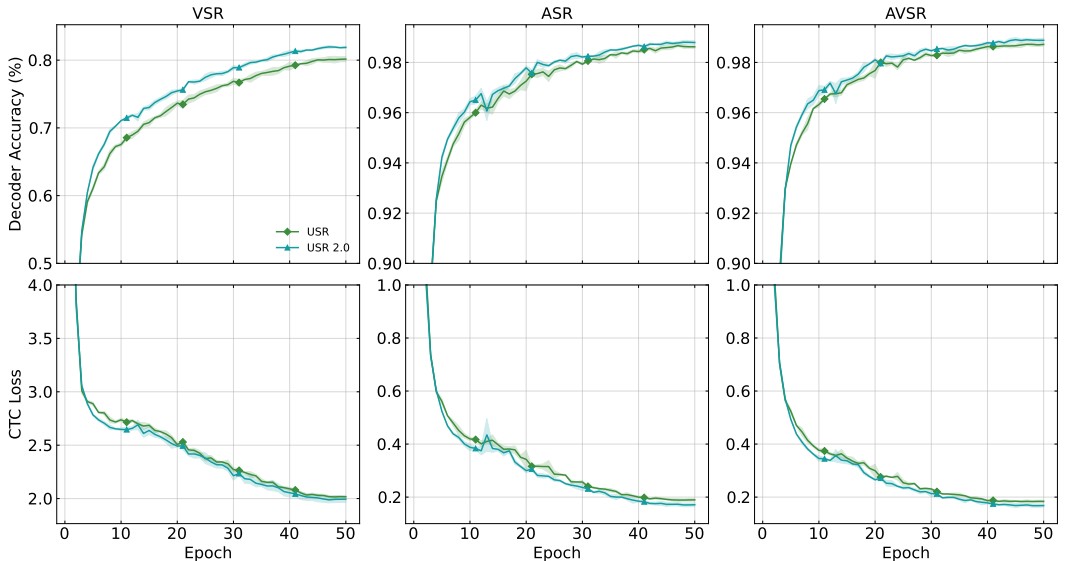

Figure 8: **Validation curves for USR and USR 2.0.** Top row shows decoder accuracy curves; bottom row shows CTC validation loss. Each curve shows mean ± std over three seeds.

## C.5    IN-DISTRIBUTION BEHAVIOUR

**Impact of decoding strategy at inference (in-distribution).**    Table 12 presents WER results for USR 2.0 under different decoding strategies on the in-distribution LRS3 low-resource benchmark. We observe that attention-only greedy decoding outperforms CTC-only decoding across ASR, VSR, and AVSR, reflecting the decoder's strong sequence modelling capabilities. As expected, the best performance is achieved using joint CTC + attention decoding with beam search (albeit at a much higher computational and memory cost). On the other hand, as we have seen in Section C.3, attention decoding tends to degrade on out-of-distribution inputs such as longer utterances, while CTC-only decoding remains stable. This contrast underscores the complementary strengths of the two branches.

We also evaluated whether CTC-driven teacher forcing can be used as an inference-time decoding strategy. As shown in Table 12, this approach yields higher WER than standard greedy or beam-search decoding. This is expected: the benefit of CTC-driven teacher forcing arises during self-training from matched teacher–student conditioning. At inference, where the model must generate coherent outputs autoregressively, CTC-driven decoding does not perform better than conventional CTC-based or AR methods.

**Validation curves and convergence.**    Figure 8 compares the validation dynamics of USR and USR 2.0 across VSR, ASR, and AVSR, using Base models trained in the low-resource setting over 50 epochs with LRS3 as the unlabelled data source (in-distribution setting). We report decoder accuracy (top row) and CTC validation loss (bottom row), averaged over three random seeds. Across all tasks, USR 2.0 consistently achieves higher decoder accuracy and lower CTC loss throughout training, indicating more effective learning within fewer total epochs.

Table 13: **Training Duration Comparison.** We report LRS3 low-resource WER (%) results for 50- and 75-epoch training. Mean and standard deviation over three random seeds are reported for the 50-epoch runs.

| Method | Epochs | V | A | AV |
|--------|--------|---|---|-----|
| USR | 50 | $37.6 \pm 0.7$ | $3.4 \pm 0.1$ | $3.2 \pm 0.1$ |
| USR 2.0 | 50 | $\mathbf{36.2} \pm 0.5$ | $\mathbf{3.0} \pm 0.1$ | $\mathbf{2.9} \pm 0.1$ |
| USR | 75 | **36.0** | 3.2 | 3.0 |
| USR 2.0 | 75 | 36.1 | **3.0** | **2.9** |

As shown in Table 13, extending training for USR to 75 epochs allows it to close much of the gap in this in-distribution setting (where even the unlabelled data is in distribution), suggesting that USR 2.0 reaches comparable or better performance with fewer epochs. This contributes to its overall training efficiency, which stems from both faster training steps and a reduced number of total steps (see Figure 4 in the main text).

## C.6 FAILURE CASES

Table 14 shows representative failure cases for the Huge USR 2.0 model on the LRS3 test set. Although the model performs strongly across modalities, some error patterns remain. First, VSR can produce visually plausible but semantically incorrect substitutions (e.g., "break" vs. "prod", "conscious" vs. "cautious"), which ASR and AVSR typically correct. Second, there are cases where ASR can drift semantically (e.g., "It wasn't a treatment" vs. "They want a treatment"), but the audiovisual model resolves the ambiguity, illustrating how visual cues can compensate for uncertain acoustics. Finally, a small number of errors persist across all modalities. These tend to be phonetically understandable (e.g., "Signfeld" vs. "Seinfeld", or "had" vs. "ought").

## D DISCUSSION / LIMITATIONS

Despite its advantages, USR 2.0 also presents limitations. First, while our proposed modifications significantly reduce training time compared to USR (nearly halving it), they do not match the efficiency of self-supervised approaches that rely on a fully supervised fine-tuning stage (Shi et al., 2022a; Haliassos et al., 2022a; 2024b). In low-resource settings, where the labelled dataset is small, such methods benefit from training the model on limited labelled data, making them substantially faster in wall-clock time. In contrast, USR and USR 2.0 iteratively optimise on both labelled and unlabelled data through self-training, leading to longer overall training durations. However, this trade-off enables unified learning across ASR, VSR, and AVSR tasks. By leveraging pseudo-labelling across modalities, USR 2.0 reduces hyperparameter sensitivity and achieves strong performance on all tasks using a single model, without sacrificing *inference* speed and with significantly lower deployment cost compared to training and maintaining separate models.

Second, we observe that the benefit of additional unlabelled data is more pronounced for VSR than for ASR or AVSR. This is likely due to the higher WERs in VSR, which allow it to benefit from increased data volume even if pseudo-labels are noisy. In contrast, ASR and AVSR, which already perform well in low-resource settings, may be bottlenecked by label quality rather than quantity. As a result, greedy decoding (used here to keep pseudo-labelling computationally feasible) may limit further improvements in ASR and AVSR, where higher-quality supervision could be more beneficial. We believe that developing strategies for improving pseudo-label quality without significantly compromising efficiency is a challenging but important area for future work.

Third, CTC-driven teacher forcing is specifically designed for iterative self-training, where the teacher evolves over time and pseudo-labels must be generated efficiently at every iteration. In this regime, matching the teacher and student conditioning is more important than global sequence coherence, and the method works well in practice. However, in non-self-training scenarios, such as offline pseudo-labelling with a frozen pre-trained model or inference-time decoding, coherence becomes desirable. In these settings, standard autoregressive or beam-search decoding is more appropriate, as they (though slower) are run only once and do not dominate training cost.

Table 14: **Representative failure cases for USR 2.0 on the LRS3 test set**. We show outputs from the Huge model. Correct fragments are shown in green; errors are in red.

| | |
|---|---|
| GT | We push and pull and poke and prod things |
| V | We push and pull and poke and break things |
| A | We push and pull and poke and prod things |
| AV | We push and pull and poke and prod things |
| GT | You're more cautious |
| V | You're more conscious |
| A | You're more cautious |
| AV | You're more cautious |
| GT | And one activist actually found a contract from a western company for the sale of surveillance |
| V | And one activist asked if I could contract from a western company for sale surveillance |
| A | And one activist actually found a contract from a western company for the sale of surveillance |
| AV | And one activist actually found a contract from a western company for the sale of surveillance |
| GT | They want a treatment |
| V | They want that treatment |
| A | It wasn't a treatment |
| AV | They want a treatment |
| GT | I said we work together |
| V | I think we work at the |
| A | I said we worked together |
| AV | I said we work together |
| GT | What difference does it make if they talk like Jerry Seinfeld |
| V | What difference does it make if they talk like church they fail |
| A | What difference does it make if they talk like Jerry Signfeld |
| AV | What difference does it make if they talk like Jerry Signfeld |
| GT | That's something we ought to celebrate |
| V | That's something we can't celebrate |
| A | That's something we had to celebrate |
| AV | That's something we had to celebrate |

## E  BROADER IMPACT

Unified speech recognition systems like USR 2.0 have the potential to expand accessibility across a wide range of real-world settings. ASR can facilitate communication for individuals with motor impairments or reading difficulties, while VSR / AVSR offer an alternative for users with aphonia or in environments with poor audio quality. Moreover, the techniques developed here may contribute to broader advances in multimodal machine learning, including audiovisual forensics. For instance, it has been shown that the ability to accurately interpret audiovisual speech can support the detection of manipulated media, such as deepfakes (Haliassos et al., 2021; 2022b).

However, this technology is not without risks. Audiovisual speech recognition may be misused for mass surveillance or covert monitoring through video feeds, particularly in contexts where individuals have not consented to such analysis. This raises important questions around privacy, informed consent, and the need for regulatory oversight. Additionally, as with most large-scale machine learning systems, performance disparities may arise if models are trained on imbalanced datasets that underrepresent certain demographic groups. This could lead to inequitable outcomes, with

worse recognition rates for users based on gender, age, accent, or ethnicity, which warrants careful consideration.

## LLM USAGE DECLARATION

In accordance with the ICLR 2026 policy on large language models (LLMs), we disclose that LLMs were used solely for polishing the language of this paper. Specifically, we employed an LLM to rephrase certain sentences for clarity and conciseness. No part of the paper's technical content, experimental design, analysis, or conclusions was generated by an LLM.

