# OpenReview forum: "Pay Attention to CTC: Fast and Robust Pseudo-Labelling for Unified Speech Recognition"
_ICLR.cc/2026/Conference — ICLR 2026 Poster_

### Official Review · Reviewer_Q3PZ · 2025-10-28

**Soundness:** 4
**Presentation:** 4
**Contribution:** 3
**Rating:** 8
**Confidence:** 4

**Summary:**

This paper introduces USR 2.0, a new semi-supervised framework for unified speech recognition (ASR, VSR, AVSR). It directly addresses two issues in the original USR: the high training cost of autoregressive (AR) pseudo-labeling and the model's brittleness on out-of-distribution (OOD) data. The core contribution is a mechanism called CTC-driven teacher forcing, where fast and robust pseudo-labels from the teacher's CTC branch are used as input prefixes to the teacher's decoder to generate attention-based targets in a single, non-autoregressive pass. This is combined with a "mixed sampling" strategy to mitigate train-test mismatch. The authors claim this approach halves training time, dramatically improves OOD robustness, and achieves state-of-the-art results with a single model.

**Strengths:**

- The paper targets a clear and significant problem: the slowness and unreliability of AR pseudo-labeling in a strong existing framework.
- The "CTC-driven teacher forcing" method is an elegant solution. It cleverly uses the CTC branch's speed and robustness to guide the attention decoder, effectively solving the speed and stability issues simultaneously.
- The claims are not just supported by standard in-distribution (ID) benchmarks but by a comprehensive suite of OOD tests, including:
   - Demonstration of clear superiority over USR on long sequences, where USR's performance collapses.
   - Shows strong performance on data with additive noise.
- The 2x speedup claim is validated with wall-clock time comparisons, showing the method is both faster per step and converges in fewer epochs .

**Weaknesses:**

- The ablations show that the 50% mixed sampling is a compromise . The pure CTC-driven mode is best for OOD robustness, while the pure AR mode is best for ID accuracy. The final method balances these, but it's a trade-off, not a solution that maximizes both simultaneously.

**Questions:**

- Could you elaborate on the mechanism that allows the decoder to learn from "incoherent" attention targets (as shown in Fig 7 ) and still produce coherent AR outputs at inference? Is it fair to characterize this as the decoder learning a mapping from a coherent CTC prefix to the teacher's (conditionally valid) next-token prediction, and that this mapping task itself is what transfers the CTC's robustness, regardless of the global coherence of the resulting target sequence?

- Figure 5 shows ID and OOD performance are optimized at opposite ends of the AR sampling probability spectrum. This suggests a simple 50/50 mix is a compromise. Did you consider a staged training approach? For example, (1) training only in the fast and robust CTC-driven mode (p=0.0) to get a robust model, and then (2) fine-tuning only in AR mode (p=1.0) to recover the last bit of in-distribution performance? This might be more effective or faster overall than mixing at every step.

- You state the 2x training speedup comes from (i) faster training steps (due to non-AR pseudo-labeling) and (ii) faster convergence (requiring 50 vs. 75 epochs). Could you provide a rough breakdown of this? How much of the 2x gain is from (i) vs. (ii)? For instance, how does the wall-clock time of a 75-epoch USR 2.0 run compare to the 75-epoch USR run?

---

> ### Author Response · Authors · 2025-11-21
>
> Thank you for your thoughtful review. We address below the key concerns raised.
>
> > The ablations show that the 50% mixed sampling is a compromise. The pure CTC-driven mode is best for OOD robustness, while the pure AR mode is best for ID accuracy. The final method balances these, but it's a trade-off, not a solution that maximizes both simultaneously.
>
> We agree that the mixed sampling probability reflects a trade-off between in-distribution accuracy and out-of-distribution robustness. We view this as an instance of a general pattern in machine learning, where models that specialise on in-distribution data often sacrifice generalisation, while those optimised for robustness may underfit in-distribution patterns. Importantly, however, as shown in Figure 5, the gap between the default setting (0.5) and the best possible values is small (ID: 2.9% vs. 2.8%, OOD: 25.0% vs. 24.2%). Moreover, this trade-off can be tuned to match different application needs, with the default setting offering a strong overall compromise. We have added a clarification on this point in Section 7.
>
> > Could you elaborate on the mechanism that allows the decoder to learn from incoherent attention targets (as shown in Fig 7) and still produce coherent AR outputs at inference? Is it fair to characterize this as the decoder learning a mapping from a coherent CTC prefix to the teacher's (conditionally valid) next-token prediction, and that this mapping task itself is what transfers the CTC's robustness, regardless of the global coherence of the resulting target sequence?
>
> The reviewer’s interpretation is exactly correct regarding the mechanism by which the decoder can learn from globally incoherent attention targets. During CTC-driven teacher forcing, the decoder learns a conditional mapping from a coherent CTC prefix to the teacher’s next-token prediction. Because inference is autoregressive and conditioned on previously generated tokens, during training the student decoder does not require global coherence in the pseudo-label output of the teacher decoder, only consistency between its prefix (input) and next-token (output).
>
> In CTC-driven mode, the robustness of CTC is transferred to the attention decoder through two mechanisms:
>
> 1. The decoder is conditioned on the CTC pseudo-labels, which are more stable under domain shift than AR-decoded pseudo-labels (Figure 1). This prevents cascaded autoregressive errors from propagating into the decoder’s input. Empirically, this is reflected in the OOD gap between rows 2 of "CTC-Driven Mode" and "AR Mode" in Table 4.
>
> 2. The decoder is trained to predict not only attention targets but also CTC tokens. This enables the decoder to learn from the inherently more robust CTC outputs when the autoregressive sequence is imperfect for domain-shifted inputs. Empirically, the effect of predicting CTC pseudo-labels in addition to attention targets can be seen by comparing rows 1 and 2 of "CTC-Driven Mode" in Table 4.
>
> We have added clarifications in Section 4.1 on the mechanism by which the decoder learns from incoherent targets, and in Section 7 on how CTC-driven robustness is transferred to the decoder.
>
> > Figure 5 shows ID and OOD performance are optimized at opposite ends of the AR sampling probability spectrum. This suggests a simple 50/50 mix is a compromise. Did you consider a staged training approach? For example, (1) training only in the fast and robust CTC-driven mode (p = 0.0) to get a robust model, and then (2) fine-tuning only in AR mode (p = 1.0) to recover the last bit of in-distribution performance? This might be more effective or faster overall than mixing at every step.
>
> We appreciate the reviewer’s suggestion. This staged strategy is closely related to the adaptive schedules we examined in Appendix C.2, where training begins predominantly in the CTC-driven mode and gradually increases the probability of AR decoding over time. Interestingly, as shown in Table 10d, we did not observe a meaningful improvement over the fixed 0.5 probability used in USR 2.0. However, we agree that such curriculum-style training could prove beneficial in other settings, and we plan to explore this further in future work.
>
> > You state the 2x training speedup comes from (i) faster training steps (due to non-AR pseudo-labeling) and (ii) faster convergence (requiring 50 vs. 75 epochs). Could you provide a rough breakdown of this? How much of the 2x gain is from (i) vs. (ii)? For instance, how does the wall-clock time of a 75-epoch USR 2.0 run compare to the 75-epoch USR run?
>
> Figure 5 (grey curve) provides the requested breakdown: at the default 0.5 sampling probability, USR 2.0 trains approximately 1.3x faster per step than USR. Thus, a 75-epoch USR 2.0 run takes around 77% of the wall-clock time of a 75-epoch USR run. The full 2x training reduction reported in the paper results from combining this per-step speedup with approximately 1.5x faster convergence (50 vs. 75 epochs).

---

> > ### Comment · Reviewer_Q3PZ · 2025-11-27
> >
> > The authors have clarified the implications of the mixed sampling strategy and also addressed the concerns regarding the decoder mechanism. However, please ensure the chronological order for tables and figures in the final version (i.e., Table 2 should come before Table 3 and Figure 4 should come before Figure 5, etc.). I'll maintain my score and recommend this paper for acceptance.

---

### Official Review · Reviewer_Qtvc · 2025-10-30

**Soundness:** 4
**Presentation:** 4
**Contribution:** 4
**Rating:** 8
**Confidence:** 4

**Summary:**

This paper introduces USR 2.0, an enhanced version of the Unified Speech Recognition (USR) framework. It effectively addresses its predecessor's limitations (high training cost, poor robustness, etc.) to deliver a more efficient and reliable semi-supervised model for ASR, VSR, and AVSR. The core innovation is CTC-driven teacher forcing, which speeds up pseudo-label generation approximately 40x and couples the two supervision signals, enabling the model to achieve new state-of-the-art results while halving the total training time.

**Strengths:**

1.	The paper itself is clearly written and easy to follow.
2.	The paper directly addresses the problems and limitations of USR.
3.	The results are robust compared to baselines.

**Weaknesses:**

There are no obvious flaws in this paper. Only a few points require clarification (see Questions).

**Questions:**

1.	Are there any failure cases for USR 2.0?
2.	Why weren’t other methods, especially USR, trained with the highest-resource setting (Huge)?
3.	Beyond speech recognition, could this CTC-driven teacher forcing paradigm be applied to other sequence-to-sequence tasks?

---

> ### Author Response · Authors · 2025-11-21
>
> Thank you for your thoughtful review. We address below the key concerns raised.
>
> > Are there any failure cases for USR 2.0?
>
> We have added a qualitative analysis of representative failure cases of USR 2.0 to the revised manuscript (see Appendix C.6 and Table 14). These examples illustrate three main patterns. First, VSR may produce visually plausible but semantically incorrect outputs (e.g., "conscious" vs. "cautious"). Second, ASR can occasionally drift semantically when acoustics are ambiguous, whereas the AVSR model seems to resolve these cases by leveraging visual cues. Finally, a small number of errors persist across all modalities and generally correspond to confusions between words that sound and look similar (e.g., "Signfeld" vs. "Seinfeld").
>
> > Why weren’t other methods, especially USR, trained with the highest-resource setting (Huge)?
>
> Training the Huge model with the full highest-resource setting is computationally intensive, and reproducing it for USR would have required greater resources than were available. Having already established the superiority of USR 2.0 over USR (and other methods) across multiple model and data scales on standard benchmarks, we focused on demonstrating that the proposed CTC-driven teacher forcing enables USR 2.0 to scale efficiently to this setting (one that exceeds the size used in prior related self- or semi-supervised works) and to achieve significantly stronger performance at that scale.
>
> > Beyond speech recognition, could this CTC-driven teacher forcing paradigm be applied to other sequence-to-sequence tasks?
>
> The proposed CTC-driven teacher forcing paradigm can apply to tasks where input and output sequences are ordered but lack explicit frame-level alignment. Examples include handwriting recognition (image of a handwritten line to text), music transcription (audio to note sequence), and DNA or protein sequencing (raw electrical signal to base sequence). We have added these examples to the Conclusion to clarify the potential broader applicability of the approach.

---

> > ### Comment · Reviewer_Qtvc · 2025-11-28
> >
> > Thank you for the insightful clarification. I have no further questions, and I will keep my positive rating of the paper. Great work!

---

### Official Review · Reviewer_t5vy · 2025-10-31

**Soundness:** 4
**Presentation:** 4
**Contribution:** 4
**Rating:** 8
**Confidence:** 4

**Summary:**

The paper proposes a framework, for semi-supervised universal speech recognition, USR 2.0 building upon USR. Hybrid CTC–attention training in USR suffers from inefficiency and instability: autoregressive (AR) decoding is slow and error-prone when generating pseudo-labels for unlabeled data, and accumulated prediction errors can degrade model quality. The authors identify that the CTC branch—though less expressive—is inherently more stable and monotonic, making it a good candidate for guiding the attention branch.

To exploit this, the paper introduces CTC-driven teacher forcing, where the teacher model’s CTC predictions are first greedily decoded and merge-collapsed and then used as fixed token prefixes to condition the attention decoder. This approach removes the need for sequential AR decoding, allowing attention-based pseudo-labels to be generated in parallel in one forward pass. Although such CTC-conditioned sequences may not be globally coherent, they still provide consistent and aligned supervision for training, since both teacher and student operate under identical conditioning.

However, this introduces a mismatch between training and inference: during training, the decoder conditions on CTC tokens, while at inference it uses its own AR outputs. To reduce this exposure bias, the paper adds a mixed-sampling strategy, randomly alternating between CTC-driven mode and standard AR mode. The CTC-driven mode emphasizes efficiency and robust alignment, while the AR mode preserves linguistic coherence and matches inference conditions. Together, these yield faster training, improved long-form stability, and better robustness to out-of-distribution data, while retaining the modeling flexibility of attention decoding.

**Strengths:**

1. CTC-driven teacher forcing is an elegant idea that leverages the stability and monotonic alignment properties of CTC to guide the attention decoder. It enables parallel pseudo-label generation without slow autoregressive decoding, thereby reducing computational cost and eliminating cascading AR errors.

2. Improved training efficiency: The paper shows a 2× reduction in training time, which is significant for multimodal setups (audio, visual, audiovisual). This is achieved without compromising accuracy, highlighting the practicality of the approach.

3. Strong robustness across conditions: The mixed-sampling strategy strikes a balance between efficiency and exposure-bias mitigation. The model demonstrates improved out-of-distribution (OOD) robustness, particularly for long-form or noisy inputs, compared to both the baseline USR and modality-specific self-supervised systems.

4. Thorough experimentation: The experiments are well-structured, spanning ASR, VSR, and AVSR tasks and several benchmarks. The ablations clearly quantify the contribution of each component.

5. Conceptual clarity and reproducibility: The paper gives a clear conceptual link between the CTC and attention paradigms, and explains the motivation for bridging them. The code and implementation details are provided, making the approach accessible to replicate.

**Weaknesses:**

The proposed approach relies heavily on the quality of CTC-generated pseudo-labels. While the paper acknowledges that the student decoder is trained to predict the teacher’s outputs under the same CTC-driven inputs, the method still assumes that these pseudo-labels are sufficiently coherent to guide decoder learning. For challenging or noisy segments, however, CTC errors can propagate through teacher forcing since the decoder conditions directly on these imperfect sequences. Consequently, the learning signal may not always be optimal, potentially limiting the broader applicability of the approach beyond the specific experimental settings demonstrated in the paper.

**Questions:**

See Weaknesses above.

---

> ### Author Response · Authors · 2025-11-21
>
> Thank you for your thoughtful review. We address below the key concern raised.
>
> > The proposed approach relies heavily on the quality of CTC-generated pseudo-labels. While the paper acknowledges that the student decoder is trained to predict the teacher’s outputs under the same CTC-driven inputs, the method still assumes that these pseudo-labels are sufficiently coherent to guide decoder learning. For challenging or noisy segments, however, CTC errors can propagate through teacher forcing since the decoder conditions directly on these imperfect sequences. Consequently, the learning signal may not always be optimal, potentially limiting the broader applicability of the approach beyond the specific experimental settings demonstrated in the paper.
>
> We agree that CTC’s conditional-independence assumptions can limit the expressiveness of the resulting pseudo-labels in challenging settings, especially on in-distribution samples where AR decoding is more powerful (Figure 1). This can introduce a form of exposure bias (as discussed in Section 4.2) and may lead to weaker attention pseudo-labels when the teacher decoder is conditioned on imperfect CTC sequences, as noted by the reviewer. In practice, however, we did not observe degradation of the decoder’s performance across the wide range of datasets and settings explored in the paper; on the contrary, we observed improved overall performance due to CTC-driven teacher forcing (Figure 5). We believe this partly stems from our mixed sampling approach, which combines CTC-driven teacher forcing and standard AR decoding (sampled with probability 0.5), balancing robustness and expressiveness during self-training. We have added a clarification in Section 4.2 to make the motivation behind this mixed sampling design more explicit.
>
> We also note that in Appendix C.2, we explored an adaptive sampling schedule, where CTC-driven teacher forcing dominates early in training (when AR decoding is more fragile) and the probability of AR pseudo-labelling increases over time (as AR decoding stabilises). As shown in Table 10d, this performs similarly to the fixed 0.5 probability used in USR 2.0, though such scheduling may yield benefits at larger scales. We plan to investigate this further in future work.

---

> > ### Comment · Reviewer_t5vy · 2025-11-27
> > **Response to Author Comments**
> >
> > Thank you for addressing the key concern, I have no further questions. I will keep my positive rating of the paper.

---

### Official Review · Reviewer_3D7N · 2025-10-31

**Soundness:** 4
**Presentation:** 3
**Contribution:** 4
**Rating:** 8
**Confidence:** 4

**Summary:**

The paper presents USR 2.0, an improved version of the Unified Speech Recognition (USR) framework that integrates ASR, VSR, and AVSR into a single model. The original USR is limited by the high cost and error accumulation of autoregressive decoding for attention-based pseudo-labels. USR 2.0 addresses this with CTC-driven teacher forcing, which uses CTC outputs to generate attention pseudo-labels in a single forward pass. A mixed sampling strategy balances the expressiveness of autoregressive training with the robustness of CTC. As a result, USR 2.0 achieves faster training, improved robustness to distribution shifts, and state-of-the-art performance on LRS2, LRS3, and WildVSR using a unified model.

**Strengths:**

* The authors' overall intuitions are reasonable and well-supported by design choices. USR 2.0 introduces a CTC-driven pseudo-labelling approach that effectively removes the autoregressive bottleneck in attention-based decoding, resulting in significantly faster training and improved inference efficiency.

* The model unifies ASR, VSR, and AVSR within a single architecture, and demonstrates strong robustness to long inputs, noise, and domain shifts, while maintaining competitive performance on ID benchmarks.

* The paper is further strengthened by a clear training strategy, a simple yet effective mixed sampling method, and a comprehensive set of experiments including ablations, OOD tests, and qualitative analysis.

**Weaknesses:**

* Since the attention decoder must operate autoregressively during inference, the benefit of test-time parallelism is limited, and the speedup primarily applies to the training phase.

* Although attention pseudo-labels generated from CTC-driven decoding may lack global coherence, the authors convincingly argue that this does not hinder learning during self-training, as both teacher and student are conditioned on the same token sequence. However, this could limit reuse of such pseudo-labels in non-self-training settings, where inference coherence is required.

* Minor weakness :
    * In Figure 2, the line representing teacher forcing looks quite similar to the other lines. Please change it to a different color for better visibility.

**Questions:**

* Mixed sampling uses AR decoding in only half of the training steps, yet inference is fully AR. Have the authors compared this to a model trained entirely with AR decoding? It would clarify how much performance is being traded for efficiency.

* Is there any potential to use CTC-driven attention decoding at inference time? While coherence may be limited, it might still be useful in latency-sensitive scenarios.

---

> ### Author Response · Authors · 2025-11-21
>
> Thank you for your thoughtful review. We address below the key concerns raised.
>
> > Since the attention decoder must operate autoregressively during inference, the benefit of test-time parallelism is limited, and the speedup primarily applies to the training phase.
>
> We agree that the proposed gains primarily target the training phase. This focus reflects a main scalability bottleneck of USR, i.e., the cost of iterative autoregressive pseudo-labelling on unlabelled data. Our method was designed to make this stage substantially more efficient, while also significantly improving robustness. We view inference-time efficiency as an interesting and complementary direction that is orthogonal to the scope of this work.
>
> > Although attention pseudo-labels generated from CTC-driven decoding may lack global coherence, the authors convincingly argue that this does not hinder learning during self-training, as both teacher and student are conditioned on the same token sequence. However, this could limit reuse of such pseudo-labels in non-self-training settings, where inference coherence is required.
>
> CTC-driven teacher forcing is designed for iterative self-training, where the randomly initialised teacher evolves throughout training and pseudo-labels must be generated efficiently at each iteration. For settings outside iterative self-training (such as inference-time decoding or offline pseudo-labelling with a frozen pre-trained model), global coherence is indeed desirable. In such cases, standard autoregressive or beam-search decoding is more appropriate, as these methods (though slower) need to be run only once and do not dominate training cost. We have expanded the discussion in Appendix D to include this point.
>
> > In Figure 2, the line representing teacher forcing looks quite similar to the other lines. Please change it to a different color for better visibility.
>
> Thank you for your suggestion. We have updated the figure to use a more distinct colour for teacher forcing.
>
> > Mixed sampling uses AR decoding in only half of the training steps, yet inference is fully AR. Have the authors compared this to a model trained entirely with AR decoding? It would clarify how much performance is being traded for efficiency.
>
> Figure 5 reports this comparison by varying the probability of sampling the AR mode. Training exclusively with AR decoding (p = 1.0) yields comparable in-distribution WER (2.9%) but substantially worse out-of-distribution performance (40.1% vs 25.0%), as autoregressive errors accumulate during pseudo-labelling. Conversely, low AR probabilities improve OOD robustness at some cost to in-distribution accuracy. The default value of 0.5 achieves a good balance, providing strong robustness and in-distribution performance, as well as faster training.
>
> > Is there any potential to use CTC-driven attention decoding at inference time? While coherence may be limited, it might still be useful in latency-sensitive scenarios.
>
> We evaluated CTC-driven attention decoding at inference, as suggested, and include the results below. As shown in the table, this approach yields higher WER than standard greedy or beam-search decoding. This outcome is consistent with our hypothesis: the benefit of CTC-driven teacher forcing comes from matched conditioning of the student/teacher decoders during self-training. In other words, CTC-driven decoding serves as an effective learning signal but does not replace CTC-based and/or autoregressive inference. We have now included this result in Table 12 and added a discussion of this point in Appendix C.5.
>
> | Decoding method | V | A | AV |
> |-----------------|----|----|----|
> | CTC-only (greedy) | 45.6 | 4.1 | 4.0 |
> | Attention-only (greedy) | 41.8 | 3.8 | 3.9 |
> | CTC + attention (beam 30) | 36.2 | 3.0 | 2.9 |
> | CTC-driven teacher forcing | 47.1 | 4.2 | 4.1 |

---

> ### Comment · Reviewer_3D7N · 2025-11-28
> **Final Recommendation**
>
> The authors have addressed my main concerns and questions in a clear and logical manner. All of my doubts have been resolved, and I have no further questions. I will maintain my positive rating. Thank you for providing such great work!

---

### Author Response · Authors · 2025-11-21

We sincerely thank all four reviewers for their thoughtful and constructive feedback. We are pleased that the reviewers consistently recognised the strength of our proposed method, the clarity and soundness of our presentation, and the breadth and rigour of our experimental evaluation.

We addressed the comments in the individual responses and updated the paper accordingly, with key changes highlighted in blue. In summary, the main revisions are:

- In Section 4.1, we expanded the explanation of how the decoder learns effectively even when attention targets are globally incoherent.
- In Section 4.2, we clarified the motivation of mixed sampling.
- In Section 7, we expanded the discussion on robustness and AR sampling probability.
- In the Conclusion, we added a note on the potential applicability of our approach for sequence-to-sequence tasks beyond speech recognition.
- In Appendix C.5, we added inference-time results using CTC-driven decoding.
- In Appendix C.6, we included failure cases for our Huge model.
- In Appendix D, we expanded the discussion on the applicability of CTC-driven forcing in non-self-training settings.

We once again thank the reviewers for their insightful comments, which helped improve the clarity and completeness of the paper. We hope the updated manuscript and per-review responses address all remaining concerns.

---

### Meta-Review · Area_Chair_S4ER · 2026-01-06

**Summary:**

This paper introduces USR 2.0, a semi-supervised framework for unified speech recognition (ASR, VSR, AVSR) that addresses key limitations of the original USR through CTC-driven teacher forcing. All four reviewers unanimously recommend acceptance (all rated 8: accept, good paper), praising the method's elegance, strong experimental validation, and significant improvements in training efficiency (2x speedup) and out-of-distribution robustness.

**Reviewer Concerns:**

**Addressed by Rebuttal:**
- The mechanism by which the decoder learns from globally incoherent attention targets was clarified, the decoder learns a conditional mapping from coherent CTC prefixes to next-token predictions, which transfers CTC's robustness regardless of global sequence coherence [Reviewer Q3PZ, Reviewer 3D7N]
- The breakdown of the 2x training speedup was provided: 1.3x faster per step plus ~1.5x faster convergence (50 vs 75 epochs) [Reviewer 3D7N]
- CTC-driven attention decoding at inference was evaluated and shown to yield higher WER than standard decoding, confirming the method's value is in training rather than inference [Reviewer Q3PZ]
- Failure cases were added in Appendix C.6 with qualitative analysis [Reviewer Qtvc]
- Broader applicability to other sequence-to-sequence tasks was discussed [Reviewer Qtvc]

Outstanding Concerns:
- The 50% mixed sampling represents a trade-off rather than a solution maximizing both ID accuracy and OOD robustness simultaneously, though the gap between default and optimal settings is small [Reviewer 3D7N]
- CTC errors can propagate through teacher forcing in challenging segments, though mixed sampling mitigates this [Reviewer t5vy]

**Reviewer Scores:**

Reviewer Q3PZ: 8. The reviewer explicitly stated they would maintain their score and recommend acceptance after the authors clarified the mixed sampling implications and decoder mechanism.

Reviewer Qtvc: 8. The reviewer confirmed no further questions remained and maintained their positive rating.

Reviewer t5vy: 8. The reviewer acknowledged the key concern was addressed and confirmed maintaining their positive rating.

Reviewer 3D7N: 8. The reviewer stated all doubts were resolved and maintained their positive rating.

---

### Decision · Program_Chairs · 2026-01-26

Accept (Poster)